# A higher order PUF complex is central to regulation of *C. elegans* germline stem cells

Chen Qiu[1], Sarah L. Crittenden[2], Brian H. Carrick [2,5], Lucas B. Dillard[3,6], Stephany J. Costa Dos Santos[2,8], Venkata P. Dandey[3,8], Robert C. Dutcher[1,8], Elizabeth G. Viverette[3,7,8], Robert N. Wine[1,8], Jennifer Woodworth[2,8], Zachary T. Campbell [4], Marvin Wickens [2], Mario J. Borgnia[3], Judith Kimble [2] ✉ & Traci M. Tanaka Hall [1] ✉

PUF RNA-binding proteins are broadly conserved stem cell regulators. Nematode PUF proteins maintain germline stem cells (GSCs) and, with key partner proteins, repress differentiation mRNAs, including *gld-1*. Here we report that PUF protein FBF-2 and its partner LST-1 form a ternary complex that represses *gld-1* via a pair of adjacent FBF binding elements (FBEs) in its 3′UTR. One LST-1 molecule links two FBF-2 molecules via motifs in the LST-1 intrinsically-disordered region; the *gld-1* FBE pair includes a well-established 'canonical' FBE and a newly-identified noncanonical FBE. Remarkably, this FBE pair drives both full RNA repression in GSCs and full RNA activation upon differentiation. Discoveries of the LST-1−FBF-2 ternary complex, the *gld-1* adjacent FBEs, and their in vivo significance predict an expanded regulatory repertoire of different assemblies of PUF-partner-RNA higher order complexes in nematode GSCs. This also suggests analogous PUF controls may await discovery in other biological contexts and organisms.

PUF (for Pumilio and FBF) RNA-binding proteins regulate gene expression post-transcriptionally throughout Eukarya. PUF proteins can regulate stability, translation and localization of target mRNAs[1–4]. To do so, they rely on extensive collaborations with protein partners. For example, *Drosophila* Pumilio works with Nanos to enhance RNA-binding affinity and impact biological processes, including germline development[5–8]. In *C. elegans*, FBF collaborates with LST-1 to enhance RNA-binding affinity[9] and maintain stem cells[10–12]. Although many PUF partners are known, most partnerships are poorly understood. A key challenge now is to learn how partnerships modulate PUF function and regulation. The *C. elegans* germline is particularly tractable for unraveling how PUF proteins maintain stem cells, a broadly conserved role

of the PUF family[13]. In the nematode, partnerships between any of four PUF proteins and either of two PUF partners promote self-renewal of germline stem cells (GSCs), albeit with extensive redundancy[14]. Here we focus on interactions between one PUF protein, FBF-2, and one partner, LST-1, and their control of RNA expression via adjacent FBF binding elements (FBEs).

The signature biochemical activity of all PUF proteins is sequence-specific binding to defined RNA regulatory elements. They do so via an RNA-binding domain (RBD) spanning eight α-helical repeats, called PUF or Pumilio repeats[1–4,13,15,16]. Prototypical PUF proteins, including *Drosophila* Pumilio and human PUM1, recognize an 8-nt RNA sequence (5′-UGUAnAUA-3′, n=any nucleotide)[3,17,18] with each repeat binding to

[1]Epigenetics and RNA Biology Laboratory, National Institute of Environmental Health Sciences, National Institutes of Health, Research Triangle Park, NC, USA. [2]Department of Biochemistry, University of Wisconsin, Madison, WI, USA. [3]Genome Integrity and Structural Biology Laboratory, National Institute of Environmental Health Sciences, National Institutes of Health, Research Triangle Park, NC, USA. [4]Department of Anesthesiology, University of Wisconsin School of Medicine and Public Health, Madison, WI, USA. [5]Present address: MRC Laboratory of Molecular Biology, Cambridge, UK. [6]Present address: Department of Biophysics and Biophysical Chemistry, Johns Hopkins University School of Medicine, Baltimore, MD, USA. [7]Present address: Department of Pharmacology and Cancer Biology, Duke University School of Medicine, Durham, NC, USA. [8]These authors contributed equally: Stephany J. Costa Dos Santos, Venkata P. Dandey, Robert C. Dutcher, Elizabeth G. Viverette, Robert N. Wine, Jennifer Woodworth. ✉e-mail: jekimble@wisc.edu; hall4@niehs.nih.gov

one nucleotide in the element[8,19]. *C. elegans* FBF-1 and FBF-2 (collectively FBF) are more flexible in the RNA sequences they recognize[20–23]. The FBF RBD (Fig. 1a) can recognize either of two core elements: a longer 'canonical' 9-nt FBE with a consensus sequence of 5'-

UGURnnAUn-3' (R=purine)[23–26] or a shorter compact 8-nt FBE (cFBE) with the consensus 5'-UGUGAA(A/U)n-3'[21,27]. In some RNAs, a cytosine upstream of either core element increases RNA-binding affinity[26,27]. For canonical FBEs, FBF repeats bind to nucleotides at the 5' and 3' ends,

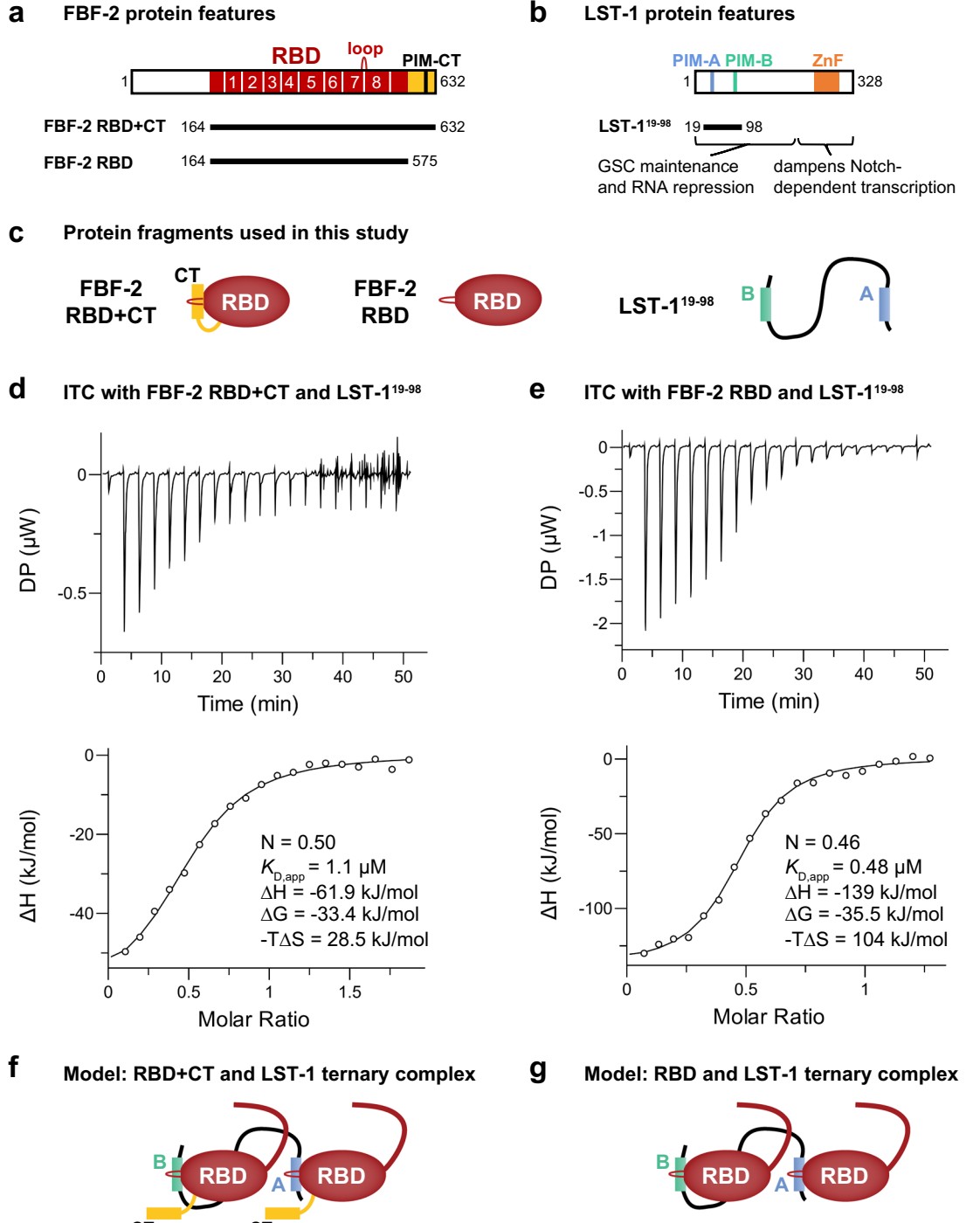

**Fig. 1 | LST-1 interacts with two FBF-2 molecules. a** FBF-2 variants used in this work. RBD (red) spans eight PUF repeats (numbered); CT (yellow); PIM-CT, PUF interacting motif in CT; loop indicates where PIMs bind. **b** LST-1 and the LST-1[19–98] fragment central to this work. PIM-A and PIM-B, LST-1 PIMs; ZnF, zinc finger. **c** Relevant protein fragments. Left, FBF-2 RBD+CT, RBD (red oval) with CT extension (yellow); middle, FBF-2 RBD only (red oval) with no CT; right, unstructured LST-1[19–98] spans PIM-A (blue) and PIM-B (green). **d,e** Above, representative ITC thermograms (differential power [DP] vs time). Below, corresponding titration curve-fitting graphs for interaction of LST-1[19–98] and FBF-2 RBD+CT (**d**) or FBF-2 RBD

(**e**). DP is the differential power to maintain a zero temperature difference between reference and sample cells. Thermodynamic parameters from one replicate indicated in lower panels; thermodynamic parameters from two distinct technical replicates are in Table 1. **f** Higher order complex with two FBF-2 and one LST-1 molecule. LST-1 links two FBF-2 RBD+CT molecules anchored by its PIMs, which displace FBF-2 CTs. The N-terminal IDR of FBF-2 is shown as a curved red line coming from the RBD. **g** Higher order complex with two FBF-2 RBDs and one LST-1 molecule.

**Table 1 | Binding affinities, stoichiometries, and thermodynamic parameters for interactions between variants of FBF-2 and LST-1**

| | FBF-2 variant | LST-1 variant [a] | wt PIM present | N (sites) [b] | $K_{D,app}$ (µM) [b] | ΔH (kJ/mol) [b] | ΔG (kJ/mol) | -TΔS (kJ/mol) |
|---|---|---|---|---|---|---|---|---|
| 1 | RBD+CT | LST-1[19–98] | A+B | 0.50 ± 0.01 | 1.1 ± 0.2 | −61.9 ± 3.6 | −33.4 | 28.5 |
| | | | | 0.57 ± 0.04 | 1.9 ± 0.9 | −45.1 ± 7.3 | −32.1 | 13.0 |
| 2 | RBD+CT | LST-1[19–98](A^m) | B | 1.02 ± 0.04 | 2.8 ± 0.8 | −47.9 ± 4.1 | −31.2 | 16.7 |
| | | | | 0.98 ± 0.02 | 3.7 ± 0.5 | −48.3 ± 2.0 | −30.5 | 17.8 |
| 3 | RBD+CT | LST-1[19–98](B^m) | A | 1.06 ± 0.25 | 24.1 ± 15.3 | −46.3 ± 21.9 | −25.9 | 20.3 |
| | | | | 0.89 ± 0.30 | 46.2 ± 24.3 | −48.3 ± 21.5 | −24.4 | 23.9 |
| 4 | RBD+CT | LST-1[19–98](A^mB^m) | none | – | Not detected | – | – | – |
| | | | | – | | – | – | – |
| 5 | RBD | LST-1[19–98] | A+B | 0.46 ± 0.01 | 0.48 ± 0.07 | −139 ± 3.7 | −35.5 | 104 |
| | | | | 0.46 ± 0.01 | 0.52 ± 0.07 | −157 ± 4.3 | −35.3 | 122 |
| 6 | RBD | LST-1[19–98](A^m) | B | 1.15 ± 0.01 | 0.13 ± 0.01 | −64.1 ± 0.7 | −38.7 | 25.4 |
| | | | | 1.05 ± 0.01 | 0.12 ± 0.03 | −68.9 ± 1.8 | −38.9 | 30.0 |
| 7 | RBD | LST-1[19–98](B^m) | A | 0.83 ± 0.02 | 1.46 ± 0.25 | −50.8 ± 1.8 | −32.8 | 18.0 |
| | | | | 0.86 ± 0.01 | 1.54 ± 0.18 | −52.5 ± 1.2 | −32.6 | 19.9 |
| 8 | RBD | LST-1[19–98](A^mB^m) | none | – | Not detected | – | – | – |
| | | | | – | | – | – | – |

[a] LST-1 A^m (K32A, L35A), LST-1 B^m (K80A, L83A, Y85A), LST-1 A^mB^m (K32A, L35A, K80A, L83A, Y85A).
[b] Two technical replicates were performed for each analysis, and values ± fitting errors are shown for each replicate.

with central nucleotides flipped away and leaving the corresponding FBF repeats unengaged[23,26].

A crucial FBF-2 partner protein is LST-1 (Fig. 1b). LST-1 is essential for GSC self-renewal, albeit redundantly with another PUF partner, SYGL-1[28]. LST-1 binds to all four self-renewal PUF proteins (FBF-1, FBF-2, PUF-3, and PUF-11), and its partnerships are critical for both GSC maintenance and RNA repression[10–12,14,28]. LST-1 has a bipartite architecture. Its N-terminal region binds to PUF proteins and is sufficient to maintain GSCs; its C-terminal region has a different role (negative feedback of Notch signaling), a function not pertinent to this work (Fig. 1b)[11,29]. The LST-1 N-terminus is an extended intrinsically-disordered region (IDR). Embedded within that unstructured region are two PUF Interacting Motifs (PIMs), called PIM-A and PIM-B, each with a Lys-xx-Leu (KxxL) sequence (Fig. 1b)[10,11,27]. Another PIM exists in the FBF-2 C-terminal tail (CT), called PIM-CT, with an SxxL sequence (Fig. 1a)[9]. Key leucines of LST-1 PIM-A and PIM-B and FBF-2 PIM-CT interact at the same place on the FBF-2 RBD: at the base of the loop between PUF repeats 7 and 8 (Fig. 1a)[9,27,30]. The FBF-2 RBD–PIM-CT intramolecular interaction brings a cluster of negatively-charged amino acid residues, also in the FBF-2 CT, near the 5′ end of FBF-2-bound RNA to decrease RNA-binding affinity[9]. The RBD–PIM-CT interaction is thus autoinhibitory. LST-1 binding to the FBF-2 RBD via either of its PIMs displaces the FBF-2 PIM-CT to increase FBF-2 RNA-binding affinity by relieving autoinhibition.

The gld-1 mRNA is a biologically critical target of FBF regulation. It was one of the first identified[24] and ranks among the highest of >1000 FBF target RNAs found with Cross Linking and Immunoprecipitation (CLIP)[21,31]. The gld-1 3′ untranslated region (3′UTR) harbors two canonical FBEs, FBEa and FBEb[24]. FBF binds more tightly to FBEa than to FBEb in vitro[23]; FBF occupancy is greater at FBEa than FBEb, and FBEa mediates stronger repression than FBEb in vivo[21,31]. FBF lowers expression of the GLD-1 protein in GSCs[24,31–33]. Because GLD-1 promotes differentiation[34–36], gld-1 repression by FBF is critical for maintaining GSCs in an undifferentiated state. LST-1 also participates in repressing gld-1 in GSCs[12,32]. In nematodes, individual PIM mutations, LST-1(A^m) or LST-1(B^m), had no apparent biological effect; only a double mutant, LST-1(A^mB^m), abolished GSC self-renewal and rendered animals sterile[11].

In this work we explore the question: if each LST-1 PIM can bind to a PUF RBD and increase its RNA-binding affinity in vitro and if a single PIM is sufficient for fertility in vivo, why does LST-1 have two PIMs? One explanation is that the LST-1 PIMs engage two PUF proteins in a ternary complex. Here we provide biochemical, structural, and biological evidence in support of this idea. We show that FBF-2 and LST-1 interact with a 2:1 stoichiometry in vitro and that assembly of this ternary complex relies on LST-1 having both its PIMs. Based on this result, we propose that LST-1 might link two FBF-2 molecules for binding to adjacent RNA elements. We identify adjacent FBEs in the gld-1 3′UTR and show that this FBE pair is required in vivo for full repression in GSCs and full activation as GSCs begin to differentiate. We observe an FBF-2–LST-1 ternary complex on an RNA with adjacent FBEs and model a structure of this quaternary complex using cryogenic-electron microscopy (cryo-EM). Bioinformatically, we identify additional adjacent FBEs among established FBF-2 target RNAs. We discuss how discovery of "higher order PUF complexes" (a ternary complex composed of two PUF proteins linked by a multivalent partner protein that binds to two RNA regulatory elements) likely expands the regulatory capability of the C. elegans "self-renewal hub" responsible for stem cell control and the possibilities for broader significance beyond nematodes.

## Results

### LST-1[19–98] and FBF-2 interact with 1:2 stoichiometry

The existence of two PIMs in LST-1 suggested that this key PUF partner protein might interact with two FBF-2 molecules. To test this possibility, we generated an LST-1 peptide (residues 19-98) spanning PIM-A and PIM-B (LST-1[19–98], Fig. 1b, c) and determined the stoichiometry and affinity of its binding to FBF-2, using Isothermal Titration Calorimetry (ITC). Purified LST-1[19–98] was titrated into an ITC cell containing FBF-2 RBD+CT, a stable fragment composed of both RBD and C-terminal tail (Fig. 1c). No RNA was included. The ITC curves appeared uniphasic (Fig. 1d), so we fit the data using a model for one set of sites (see Methods). The one-site data fitting model revealed an N of 0.5 (Fig. 1d, lower graph; Table 1, line 1), reflecting a 1:2 stoichiometry with one LST-1 molecule binding to two FBF-2 molecules. We also determined an overall $K_{D,app}$ of 1.1 µM for the LST-1[19–98] interaction with FBF-2 RBD+CT as well as ΔG, ΔH and -TΔS values of the interaction (Fig. 1d, lower graph; Table 1, line 1).

We next investigated the interaction of LST-1[19–98] with the FBF-2 RBD lacking its CT (Fig. 1c). Similar to binding to RBD+CT, N was 0.46 (Fig. 1e, lower graph; Table 1, line 5), indicating that LST-1[19–98] binds to two FBF-2 RBD molecules. Comparison of the data for binding of LST-1[19–98] to FBF-2 RBD vs FBF-2 RBD+CT revealed a few differences (Table 1, compare lines 1 and 5). LST-1[19–98] bound to FBF-2 RBD+CT with a 2-fold

lower affinity than to FBF-2 RBD. The lower affinity for LST-1 binding to RBD+CT was also observed for shorter LST-1 peptides including only PIM-A or PIM-B and is likely due to competition with the FBF-2 PIM-CT[9]. Thermodynamic parameters provide additional insights into the binding affinities. Enthalpy for binding of LST-1[19–98] to RBD alone was more favorable than to RBD+CT, which we attribute to energy required to displace the CT. Conversely, entropy for binding of LST-1[19–98] to RBD alone was less favorable than to RBD+CT, which we attribute to increased entropy with release of the CT. Nonetheless, in the presence or absence of the CT, one LST-1[19–98] binds to two FBF-2 RBDs to form a ternary complex with 1:2 stoichiometry (Fig. 1f, g).

## Two LST-1 PIMs enable ternary complex formation

The presence of two PIMs in LST-1 together with the 1:2 LST-1:FBF-2 stoichiometry suggested that each LST-1 PIM binds a single FBF-2 molecule. We therefore asked how each LST-1 PIM contributes to FBF-2 binding affinity and stoichiometry of the complex. We first identified mutations in each PIM that disrupt binding to FBF-2. Combining results from mutations tested previously in yeast and nematodes and interactions observed in crystal structures[11,27,30], a K32A/L35A double mutant in PIM-A and a K80A/L83A/Y85A triple mutant in PIM-B disrupted their binding to FBF-2. Peptides carrying these mutants are referred to as LST-1[19–98](A^m) and LST-1[19–98](B^m) respectively. LST-1[19–98](A^m) retains an intact PIM-B and assays PIM-B function. Similarly, LST-1[19–98](B^m) harbors an intact PIM-A and assays PIM-A function.

With only one PIM intact, either PIM-A or PIM-B, the LST-1:FBF-2 stoichiometry changed to ~1:1 (Supplementary Fig. 1). This applied to both FBF-2 RBD+CT (Supplementary Fig. 1a, b) and the RBD (Supplementary Fig. 1d, e) (Table 1, lines 2,3,6,7). We previously demonstrated with shorter peptides containing a single LST-1 PIM that PIM-B binds with higher affinity to FBF-2 than PIM-A, and this tighter binding of PIM-B is due to amino acid residues flanking the KxxL motifs[30]. In the context of the longer peptides, PIM-B within LST-1[19–98](A^m) bound to RBD+CT with higher affinity than PIM-A within LST-1[19–98](B^m) (Table 1, lines 2 and 3; Supplementary Fig. 1a, b); a similar difference was seen for binding to FBF-2 RBD alone (Table 1, lines 6 and 7; Supplementary Fig. 1d, e). Both LST-1 peptides with one intact PIM had a weaker affinity for RBD+CT than for RBD, likely due to competition with the FBF-2 PIM-CT. Mutation of both LST-1 PIMs, LST-1[19–98](A^mB^m), eliminated interactions with both RBD and RBD+CT (Table 1, lines 4 and 8; Supplementary Fig. 1c, f), as expected. We conclude that formation of a ternary complex requires two LST-1 PIMs (Fig. 1f, g).

## *gld-1* FBEa*, a non-canonical element adjacent to FBEa

Given that LST-1 links two FBF-2 molecules (Fig. 1f, g), we reasoned that LST-1 and FBF-2 might regulate mRNAs harboring two adjacent FBEs. The *gld-1* mRNA 3′UTR contains two canonical FBEs, FBEa and FBEb, separated by 388 nucleotides (Fig. 2a). We suspected that two closer FBEs might exist within this major target and explored sequences near the stronger FBEa site with that in mind. A candidate emerged just three nucleotides downstream of FBEa (Fig. 2a). Its sequence (5′-**CAUGU**U**GCCAU**U-3′) contained key FBE features (boldface): a 5′ UGU, 3′ AU, and an upstream C, and these features are conserved among *Caenorhabditids* (Fig. 2b). The candidate sequence deviated from a canonical FBE: a uridine followed the UGU, rather than typical purine, and a second uridine replaced the typical final adenosine. Nonetheless, the candidate sequence was related to FBEa (5′-**CAUGU**GCC**AU**A-3′), and FBF CLIP revealed a broad peak over *gld-1* FBEa that could cover two adjacent FBEs (Fig. 2a)[21,31]. We dub this candidate element *gld-1* FBEa* and refer to the adjacent sites as *gld-1* FBEa-FBEa*.

To ask whether FBF-2 recognizes FBEa*, we took both structural and biochemical approaches. We first determined a 2.2 Å resolution crystal structure of the FBF-2 RBD in complex with an 11-nt RNA

carrying FBEa* (Supplementary Table 1). Indeed, FBF-2 bound to FBEa* by recognizing the 5′ and 3′ nucleotides common to FBEa and FBEa*: FBF-2 interacts with 5′-CAUGU and CCAU-3′ in both FBEa (Fig. 2c) and FBEa* (Fig. 2d). The FBF-2 RBD structure changes little (Supplementary Fig. 2a, RMSD = 0.37 Å over 2,744 atoms). The RNA sequences align in the 5′ and 3′ regions, and the side chains that contact the RNA bases in each structure are equivalent and positioned similarly (Supplementary Fig. 2a-c). No electron density was seen for the atypical U4 base of FBEa* (Fig. 2e, cyan arrow), so U4 is likely flipped away from the RNA-binding surface, as seen with other PUF binding elements with flipped bases[37,38]. Following the flipped U4 nucleotide, PUF repeat 5 (R5) interacted with G5 of FBEa* in a manner similar to its interaction with G4 of FBEa (Fig. 2c, d)[23].

We next measured FBF-2 binding affinities to a 14-nt RNA carrying FBEa* using an electrophoretic mobility shift assay (EMSA). Despite equivalent contacts with the two RNAs, FBF-2 RBD bound to FBEa* RNA with a $K_D = 427$ nM (Fig. 2f, line 1; Supplementary Fig. 2d), ~6-fold weaker than its previously reported binding to FBEa RNA ($K_D = 70$ nM)[9]. The difference in binding affinity may therefore be due to other factors such as additional energy needed to flip the U4 base of FBEa* or destabilization of binding caused by the flipped base. As expected due to CT inhibition, FBF-2 RBD+CT bound to FBEa* with a much lower affinity ($K_D > 2500$ nM, Fig. 2f, line 2; Supplementary Fig. 2e). The FBF-2 PIM-CT employs a key L610 residue for RBD binding and autoinhibiting RNA-binding affinity[9]. Substitution of this key leucine with alanine (L610A) eliminated autoinhibition and increased RNA-binding affinity to FBEa* ($K_D = 385$ nM, Fig. 2f, line 3; Supplementary Fig. 2f).

To test for effects of LST-1 on FBF-2 binding to FBEa*, we added either of two short peptides, each carrying an LST-1 PIM, LST-1[19–98] with PIM-A or LST-1[67–98] with PIM-B. LST-1[19–98] had little or no effect ($K_D > 2500$ nM, Fig. 2f, lines 2 and 4; Supplementary Fig. 2e, g), likely due to the weak PIM-A interaction with FBF-2 RBD+CT[9]. By contrast, LST-1[67–98] strengthened affinity of FBF-2 RBD+CT for FBEa* RNA (from $K_D > 2500$ nM to $K_D = 300$ nM, Fig. 2f, lines 2 and 5; Supplementary Fig. 2e, h), similar to the effect of LST-1 on binding to FBEa RNA[9]. In sum, both structural and in vitro binding data demonstrate that FBF-2 can bind to the newly found non-canonical *gld-1* FBEa*, and in vitro binding data shows that LST-1 and the FBF-2 CT can both affect that interaction.

## *gld-1* FBEa* is a functional regulatory element in nematodes

To ask whether *gld-1* FBEa* is a functional regulatory element in living animals, we generated FBEa* mutants (FBEa*^m) at the endogenous *gld-1* locus with CRISPR/Cas9 gene editing (Fig. 3a). Like wild-type animals, FBEa* mutants were fertile; their germlines had a normal architecture maintaining GSCs and switching from spermatogenesis to oogenesis (Fig. 3b; Fig. 3c, lines 1 and 2). We next mutated the adjacent *gld-1* FBEa element in an FBEa* mutant, and asked whether the phenotype changed. Whereas single mutants of either FBEa* or FBEa were nearly all fertile (Fig. 3c, lines 2 and 3), most FBEa FBEa* double mutants were sterile due to failure of the sperm/oocyte switch (Fig. 3c, line 5). Failure of the sperm/oocyte switch was also seen when both canonical elements, FBEa and FBEb, were mutated (Fig. 3c, line 7)[31]. This phenotypic change demonstrates that FBEa* is a functional element in nematodes and suggests that FBEa and FBEa* work together as a pair of adjacent RNA elements.

To explore the molecular function of FBEa*, we quantitated GLD-1 protein that was made from *gld-1* RNAs harboring an FBEa* mutant. This was done as a function of position in the distal gonad, where germ cells develop from self-renewal distally to overt differentiation proximally (Fig. 3d, above). FBF-2 is present throughout the region, but LST-1 is restricted to GSCs at the distal end (Fig. 3d, above)[12,31]. In wild-type germlines, GLD-1 protein is barely detectable in GSCs, increases steadily as germ cells mature, and peaks upon

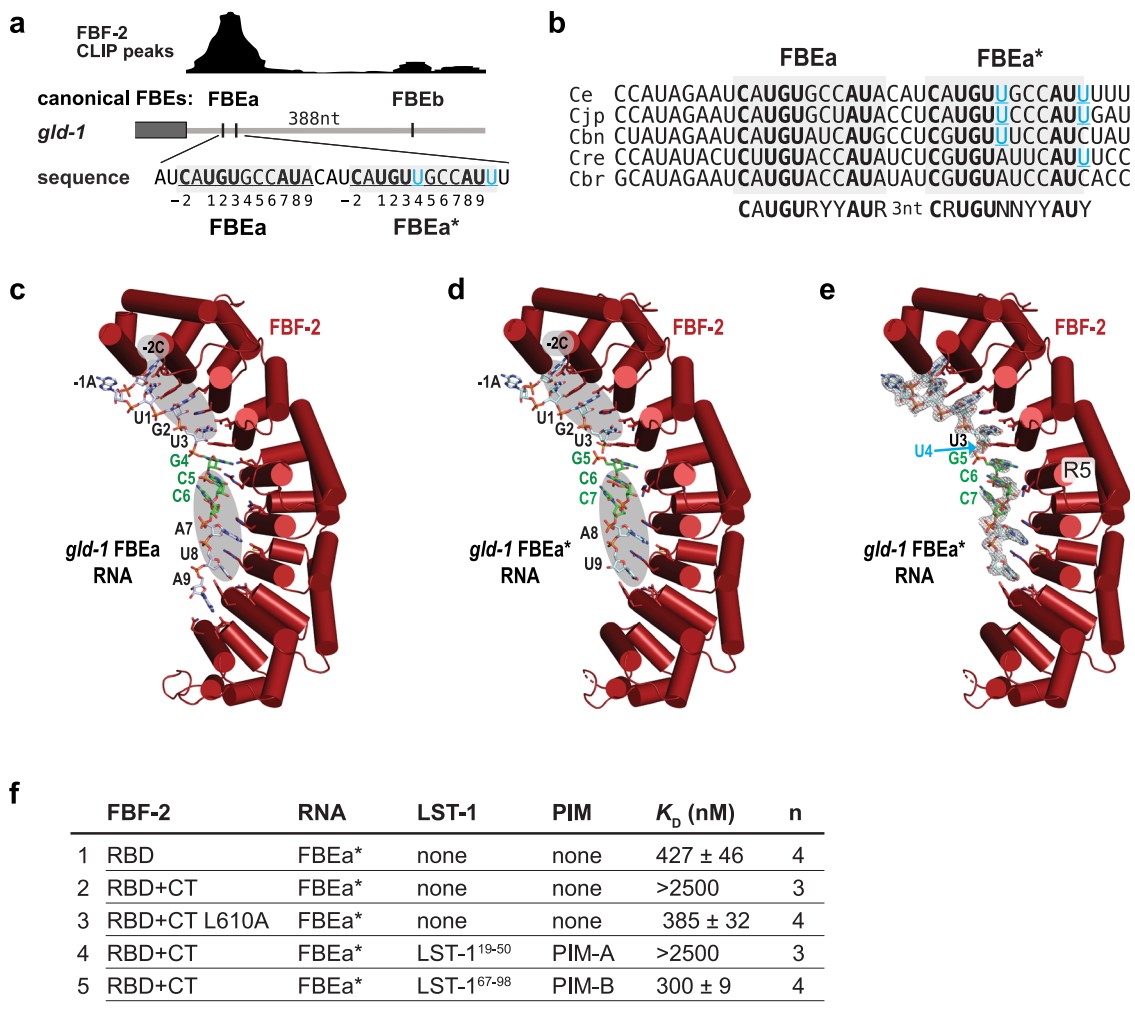

**Fig. 2 | FBF-2 RBD recognizes conserved sequence features in *gld-1* FBEa and FBEa\*. a** FBEa\* is a non-canonical FBE adjacent to FBEa in the *gld-1* 3′UTR. Gray rectangle, coding region; gray line, 3′UTR; vertical black lines, FBEs. Above 3′UTR, CLIP peaks[31] of FBF-2 binding. Larger peak includes FBEa and FBEa\*; smaller peak includes FBEb. Below 3′UTR, wild-type sequences of FBEa and FBEa\* (underlined and shaded gray) with nucleotides numbered according to convention. **b** Conservation in sequence and spacing of FBEa and FBEa\* in *Caenorhabditid* species (Ce, *C. elegans*; Cjp, *C. japonicum*; Cbn, *C. brenneri*; Cre, *C. remanei*; Cbr, *C. briggsae*). **c**–**e** Ribbon diagrams of FBF-2 RBD with cylindrical helices (red) in complex with FBE RNAs (stick models). Gray ovals mark shared 5′-CAUGU and CCAU-3′ nucleotides in FBEa and FBEa\* (**c,d**). Central GCC sequences that are stacked and flipped away from the RNA-binding surface are colored green. Although these nucleotides are not conserved among FBEs, they are shared between FBEa and FBEa\*. Nucleotide numbering corresponds to numbers below sequence in A. (**c**) Crystal structure of FBF-2 RBD bound to *gld-1* FBEa RNA (PDB ID: 3V74)[26]. (**d**) Crystal structure of FBF-2 RBD bound to *gld-1* FBEa\* RNA (5′-CAU-GUUGCCAU-3′). (**e**) Crystal structure of FBF-2 RBD bound to *gld-1* FBEa\* with composite 2F₀-F𝒸 omit map contoured at 1 σ superimposed on the FBEa\* RNA. The position of the disordered, and therefore presumably flipped, U4 in FBEa\* is indicated by a cyan arrow. **f** RNA-binding affinities to FBEa\* measured by EMSA. Mean $K_D$ values and standard error of the mean (SEM) from 'n' distinct technical replicates are reported.

differentiation (entry into meiotic prophase) (Fig. 3d, middle and below), as shown previously[32]. This dynamic GLD-1 pattern is dependent on FBF binding to FBEs in the *gld-1* 3′UTR. FBF together with a repressive partner (e.g. LST-1) lowers GLD-1 expression in GSCs[24,31–33], but more proximally, FBF switches to an activating mode, likely with an activating partner (e.g GLD-2)[25,31,32,39]. An FBE can therefore anchor either a repressive or activating FBF complex, depending on germline position. Mutants defective for repression will generate more GLD-1 than normal, while mutants defective for activation will generate less GLD-1 than normal.

Comparison of GLD-1 levels in wild-type animals and FBEa\* mutants revealed that the FBEa\* element is critical for establishing the normal *gld-1* pattern (Fig. 3e,h,i, Supplementary Fig. 3a,b, Supplementary Table 2). FBEa\* mutants had more GLD-1 than normal in GSCs (Fig. 3e, upward arrow; Fig. 3c, line 2), and about the same level proximally. A similar result was found in an FBEa mutant though the effect on repression was more severe (Fig. 3f, upward arrow; Fig. 3c,

line 3)[31]. A double mutation of both FBEa and FBEa\* in the same 3′UTR affected repression in GSCs (Fig. 3h, upward arrow; Fig. 3c, line 5), similar to the effect in FBEa mutants, but the double mutant generated significantly less GLD-1 proximally (Fig. 3h, downward arrow; Fig. 3c, line 5), suggesting synergy between these two elements for activation. One caveat might have been that FBEa FBEa\* double mutants are sterile. To exclude the possibility that this phenotypic difference might explain that synergy, we generated the same double mutant in the *gld-1* 3′UTR of a GFP reporter transgene. These reporter FBE mutants were fertile, and yet their effects were comparable to the corresponding FBE mutations at the endogenous locus (Supplementary Fig. 3b,c). One concern might have been that lowered proximal GLD-1 represents a downstream consequence of distal derepression rather than a defect in proximal activation. However, distal derepression is not coupled to proximal deactivation (Fig. 3e, f,i). Regardless, we conclude that FBEa\* is a functional element and that it works with FBEa to maintain a normal GLD-1 pattern in the distal germline.

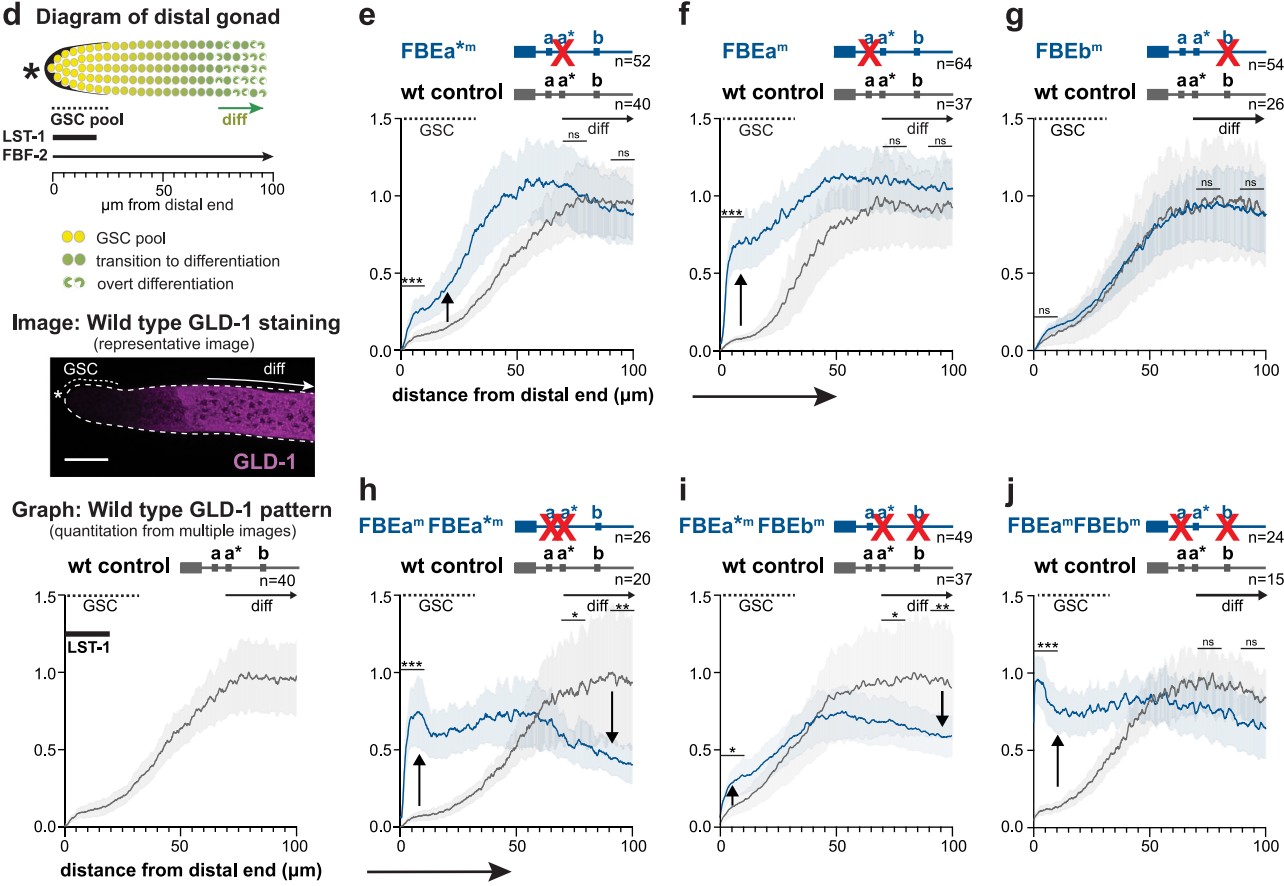

**a** wild-type AU**CAUGUGCCAUA**CAU**CAUGUUGCCAUU**U...//...AUUUUUU**CUGUGUUAU**CUUAA
edits AUCA ACAGCG AUACAUCA ACAUGCCAUUU...//...AUUUUUUC ACAGUUAUCUUAU
FBEa FBEa* FBEb

**c**

| Genotype FBE | Phenotype % Fertile | FBE state WT/mutant | 0-10 µm from distal end | Repression | 90-100 µm from distal end | Activation | Citation |
|---|---|---|---|---|---|---|---|
| 1 Wild-type | 100 | a a* b | + | full | +++++ | full | Carrick et al[31] |
| **Singles** | | | | | | | |
| 2 FBEa*m | 100 | a a* b | ++ | partial | +++++ | full | This work |
| 3 FBEam | 97 | a a* b | +++ | partial | +++++ | full | Carrick et al[31] |
| 4 FBEbm | 100 | a a* b | + | full | +++++ | full | Carrick et al[31] |
| **Doubles** | | | | | | | |
| 5 FBEam FBEa*m | 13 | a a* b | +++ | partial | +++ | partial | This work |
| 6 FBEa*m FBEbm | 100 | a a* b | ++ | partial | +++ | partial | This work |
| 7 FBEam FBEbm | 13 | a a* b | +++++ | none | ++++ | partial | Carrick et al[31] |

GLD-1 in specific regions[1] (levels compared to wild-type peak)

[1] Levels relative to peak GLD-1 in wild-type animals, 90-100µm from distal end: +++++, ~100%; ++++, ~80%; +++, ~50%; ++, ~20%; +,~10%

Given the three FBEs (FBEa, FBEa*, FBEb) in the *gld-1* 3'UTR, we asked whether any one of the FBE elements was sufficient for fertility or GLD-1 regulation by examining additional FBE double mutants. For example, wild-type FBEa is retained in double mutants defective in FBEa* and FBEb. That FBEa* FBEb double mutant was fertile (Fig. 3c, line 6), but GLD-1 was higher than normal in GSCs and lower than normal proximally (Fig. 3i, upward and downward arrows; Fig. 3c, line 6). The other FBE double mutants, retaining wild-type FBEb or FBEa* elements, respectively, were mostly sterile and also had GLD-1 pattern defects (Fig. 3c, line 5 and 7; Fig. 3h,j).

Therefore, only FBEa was sufficient for fertility and none was sufficient for normal GLD-1 regulation. We conclude that two FBEs are optimal for the GLD-1 pattern, consistent with FBEs working together.

The GLD-1 expression patterns in FBEa and FBEb single mutants provide additional insights into which FBEs work as pairs for regulation of the GLD-1 pattern. In FBEa mutants, the remaining intact elements, FBEa* and FBEb, can work together for activation but not repression (Fig. 3c, line 3; Fig. 3f), and in FBEb mutants, FBEa* can work with FBEa for both repression and activation (Fig. 3c, line 4; Fig. 3g).

**Fig. 3 | FBEa\* is a functional element. a** FBEa, FBEa\*, and FBEb sequences (underlined and shaded gray). Red, mutated; black boldface, common to FBEa and FBEa\*; cyan, FBEa\* only. **b** *C. elegans* gonad. Above, hermaphrodite with two U-shaped gonadal arms, oocytes (pink), sperm (blue). Below, gonadal arm, distal end (\*). **c** Key features of FBE mutants. Left, phenotypes: %Fertile, all sterile animals lost sperm/oocyte (s/o) switch and some lacked GSCs: FBEa[m], 1% no GSCs, n = 700; FBEa[m] FBEa\*[m], 16% no GSCs, *n* = 68; FBEa[m] FBEb[m], 7% no GSCs, n = 88. Right, GLD-1 levels. See also Supplementary Fig. 3. **d** Above, distal gonad (boxed in Fig. 3b) with germ cell states: GSCs in mitotic cell cycle (yellow), germ cells transitioning into meiotic cell cycle (green circles), and differentiated (diff) germ cells in meiotic prophase (green crescents). Shown below: extents of GSC pool (dotted line), diff cells (green arrow), LST-1 (thick black line)[11], FBF-2 (black arrow)[31] and μm measurement from distal end. Middle, representative image of distal gonad stained for GLD-1. Scale bar (bottom left), 20 μm. Below, quantitation of wild-type GLD-1

pattern as a function of position in the distal gonad, as reported[31,32]. Wild-type and mutant data obtained from 'n' gonads processed and imaged together. Annotation as in Fig. 3e-j. Gray line, mean abundance; shading, 95% confidence interval. **e–j** GLD-1 abundance in FBE mutants compared to wild-type. Gray line, mean abundance in wild-type; blue line, mean abundance in FBE mutant; shading, 95% confidence interval. p-values using a two-sided unpaired t-test assuming equal variance (no corrections for multiple comparisons) assessed using pooled data from three regions (0–10 μm, 70–80 μm and 90–100 μm) as follows: \*\*\**p* < 0.001, \*\**p* < 0.01, \**p* < 0.05, ns (not significant) *p* > 0.05. Supplementary Table 2, exact p-values; Supplementary Fig. 3a. representative images. Wild-type, FBEa\*[m] and FBEa[m] FBEa\*[m] (three replicates); FBEa\*[m] FBEb[m] (two replicates); FBEa[m], FBEb[m], FBEa[m] FBEb[m] details in ref. [31]. Upward arrow, GLD-1 > wild-type; downward arrow, GLD-1 <wild-type. **e** FBEa\*[m]. **f** FBEa[m] (modified from[31]). **g** FBEb[m] (modified from[31]). **h** FBEa[m] FBEa\*[m]. **i** FBEa\*[m] FBEb[m]. **j** FBEa[m] FBEb[m] (modified from[31]).

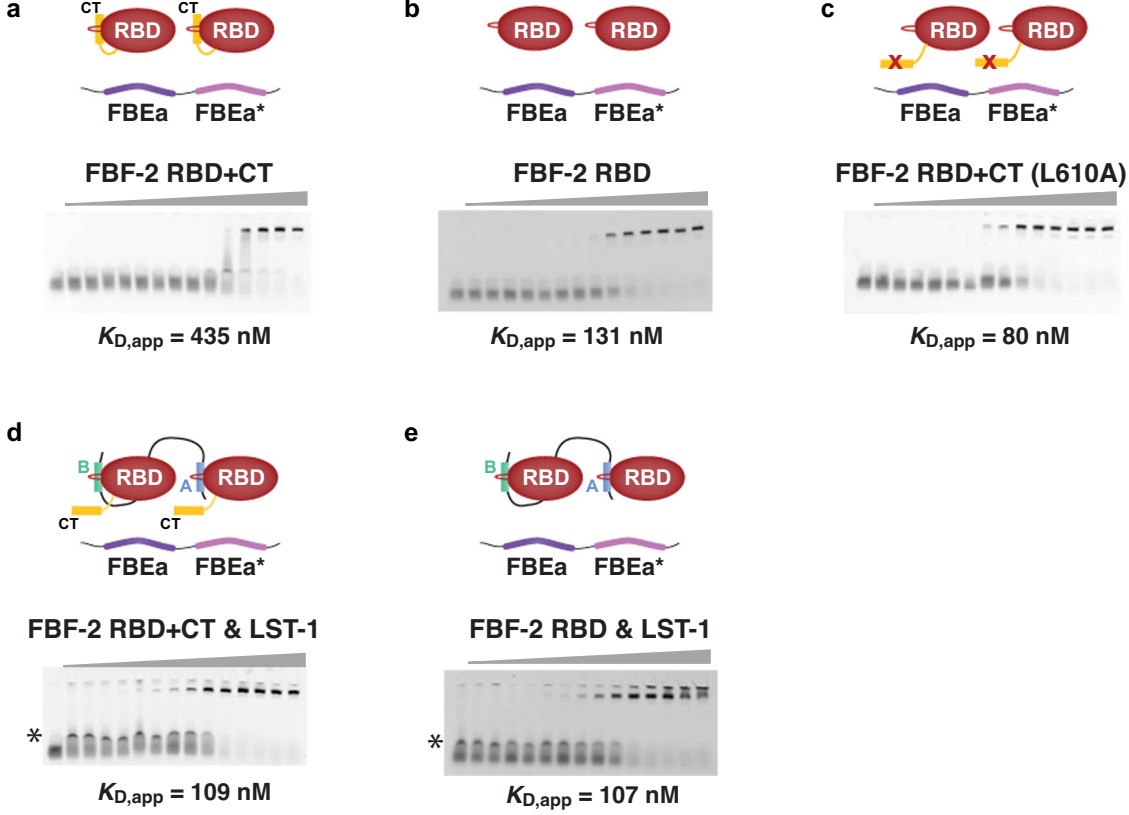

**Fig. 4 | LST-1[19–98] increases the RNA-binding affinity of FBF-2 RBD+CT.** Representative EMSA gels are shown for binding to FBEa-FBEa\* RNA by **a** FBF-2 RBD+CT, **b** FBF-2 RBD, **c** FBF-2 RBD+CT L610A, **d** FBF-2 RBD+CT with 10 μM LST-1[19–98], and **e** FBF-2 RBD with 10 μM LST-1[19–98]. Experimental components indicated in diagrams above gels. In panels d and e, an intermediate band (\*) is non-specific interaction of LST-1[19–98] with the RNA. Similar bands were detected previously with short LST-1 peptides[9], and with LST-1[19–98] and mutated FBEa-FBEa\* RNA (see source data). Mean $K_{D,app}$ values from at least three distinct technical replicates are reported. See also Table 2 and Source Data.

In sum, individual FBEs can function on their own, but they also work jointly. In GSCs, FBEa and FBEa\* function as a pair to drive full repression, whereas in differentiating germ cells, FBEa\* can work with either the adjacent FBEa or the more distant FBEb to drive full activation. In each case, we define "full" as equivalent to the wild-type level of expression. These findings support the model that FBEa and FBEa\* recruit a higher order complex for repression and suggest that any two of the other FBEs recruit a different higher order complex for activation (see Discussion).

**Synergistic binding of FBF-2 RBD+CT to *gld-1* FBEa-FBEa\* RNA**
Given that *gld-1* FBEa and FBEa\* work together in vivo, we next asked how FBEa\* affects FBF-2 binding to RNA in vitro. We first measured binding of both FBF-2 RBD+CT and FBF-2 RBD without its CT to a 29-nt

*gld-1* RNA fragment carrying the tandem FBEa and FBEa\* sites (FBEa-FBEa\* RNA) by EMSA (Fig. 4, Table 2). FBF-2 RBD+CT bound to the FBEa-FBEa\* RNA with an apparent $K_D$ ($K_{D,app}$) of 435 nM (Fig. 4a), while FBF-2 RBD bound with higher affinity ($K_{D,app}$ = 131 nM) (Fig. 4b). This difference was expected since FBF-2 CT lowers RNA-binding affinity[9]. Consistent with that explanation, FBF-2 RBD+CT carrying an L610A mutation, which disrupts the RBD–CT interaction, bound with higher affinity ($K_{D,app}$ = 80 nM) (Fig. 4c). These overall binding affinities are similar to those for FBF-2 RBD+CT (334 nM), RBD (70 nM), and L610A (82 nM) binding to a 14-nt FBEa RNA[9].

To probe the contributions of individual FBEa and FBEa\* sites to FBF-2 RNA binding, we substituted the UGU trinucleotide with ACA in each site of the FBEa-FBEa\* RNA. We thus generated three additional RNAs to be tested: (1) FBEa-FBEa\*[m] with an intact FBEa, (2) FBEa[m]-

**Table 2 | RNA-binding affinities measured by EMSA[1]**

| FBF-2 | RNA | FBEsintact | LST-1 | Replicatesn | $K_{D,app} \pm$ SEM (nM) |
|---|---|---|---|---|---|
| RBD+CT | FBEa-FBEa* | a a* | none | 3 | 435 ± 25 |
| RBD | FBEa-FBEa* | a a* | none | 4 | 131 ± 10 |
| RBD+CT L610A | FBEa-FBEa* | a a* | none | 4 | 80 ± 11 |
| RBD+CT | FBEa-FBEa*m | a | none | 3 | 1190 ± 9 |
| RBD+CT | FBEam-FBEa* | a* | none | 3 | >5000 |
| RBD+CT | FBEam-FBEa*m | none | none | 3 | none |
| RBD+CT | FBEa-FBEa* | a a* | 19-98 | 3 | 109 ± 2 |
| RBD | FBEa-FBEa*m | a | none | 4 | 145 ± 4 |
| RBD | FBEam-FBEa* | a* | none | 3 | 323 ± 20 |
| RBD | FBEam-FBEa*m | none | none | 3 | >2500 |
| RBD | FBEa-FBEa* | a a* | 19-98 | 3 | 107 ± 3 |

[1]See Supplementary Table 4 for RNA sequences.

FBEa* with an intact FBEa*, and (3) FBEa^m-FBEa*^m with no intact FBE. We again tested both FBF-2 RBD+CT and FBF-2 RBD for RNA binding by EMSA (Supplementary Fig. 4, Table 2). FBF-2 RBD+CT bound more weakly to FBEa-FBEa*^m RNA ($K_{D,app}$ = 1190 nM) than to wild-type FBEa-FBEa* RNA ($K_{D,app}$ = 435 nM) (Supplementary Fig. 4a), and essentially did not bind to FBEa^m-FBEa* RNA ($K_{D,app}$ > 5000 nM) (Supplementary Fig. 4b). The lack of FBF-2 RBD+CT binding to FBEa^m-FBEa* RNA is consistent with its lack of binding to a short RNA that carries only FBEa* (Supplementary Fig. 2e). FBF-2 RBD+CT also failed to bind to FBEa^m-FBEa*^m, which lacks both sites (Supplementary Fig. 4c). Thus, binding affinity to the wild-type FBEa-FBEa* RNA was stronger than binding to either RNA with one intact site. We conclude that FBEa* strengthens FBF-2 RBD+CT binding to RNA and suggest that the two sites act synergistically.

In contrast to FBF-2 RBD+CT, FBF-2 RBD binding to wild-type FBEa-FBEa* and the singly mutated RNAs was similar ($K_{D,app}$ = 131 nM for wild-type, 145 nM for FBEa-FBEa*^m, and 323 nM for FBEa^m-FBEa*) (Supplementary Fig. 4d–f; Table 2), but undetectable to the double mutated RNA. We suggest that the weaker binding due to autoinhibition by the FBF-2 CT allows detection of FBEa-FBEa* synergy with RBD+CT, while the stronger binding without the CT masks that synergy with the RBD alone. Regardless, RBD+CT synergistic binding is likely relevant as the CT is present in wild-type FBF-2.

We next explored the effect of LST-1^19–98 on binding of FBF-2 RBD+CT to FBEa-FBEa* RNA by EMSA. Addition of LST-1^19–98 increased RNA-binding affinity 4-fold ($K_{D,app}$ = 109 nM with LST-1 vs 435 nM without LST-1, Fig. 4a, d; Table 2). In contrast, LST-1^19–98 did not change RNA-binding affinity of the RBD alone ($K_{D,app}$ = 131 nM vs 107 nM, Fig. 4b, e; Table 2). We conclude that the FBF-2 CT is autoinhibitory for binding to two adjacent FBEs, much like it is for binding to one FBE[9], and we also conclude that LST-1 relieves that autoinhibition.

## Cryo-EM visualizes a higher-order complex of FBF-2/LST-1/RNA

We used single-particle cryo-EM to generate molecular models of the predicted FBF-2/LST-1/RNA quaternary complex. For these experiments, we prepared complexes using a 29-nt *gld-1* FBEa-FBEa* RNA, LST-1^19–98, and either FBF-2 RBD+CT or RBD. We first characterized complexes with Size Exclusion Chromatography-Multi-Angle Light Scattering (SEC-MALS). For both FBF-2 RBD+CT and the RBD, a major peak corresponding to a stable quaternary complex matched the expected molecular weight for a 2:1:1 complex of FBF-2/LST-1^19–98/FBEa-FBEa* (Supplementary Fig. 5a, b). We then prepared cryo-EM grids with the purified quaternary complexes. Pilot screening of complexes with FBF-2 RBD+CT or RBD yielded 2D classes that appeared similar, but 3D reconstructions for complexes with RBD+CT were lower resolution

than those with RBD. We therefore focused on FBF-2 RBD-containing complexes.

We collected data for FBF-2 RBD/ LST-1^19–98/ FBEa-FBEa* grids on a Talos Arctica electron microscope operated at 200 keV (Supplementary Fig. 5c, Supplementary Table 3). After 2D classification, crescent shapes characteristic of PUF RBDs were clearly visible (Fig. 5a, Supplementary Fig. 5d). Among the 2D classes, we observed two FBF-2 molecules in ~50% of the particles, and a solitary FBF-2 molecule in the other half of particles due to dissociation of the quaternary complexes, most likely caused by interaction with the air-water interface during sample vitrification (Supplementary Fig. 6a, iii). After ab initio reconstruction and refinement in CryoSparc[40], we obtained two maps: a map of a single FBF-2 molecule displaying rod-like densities for individual helices and densities for RNA bases at 4.4 Å resolution (Fig. 5b; Supplementary Fig. 6a, v-1, b, c) and a map with two well-resolved FBF-2 molecules with apparent densities for RNA bases at 6.4 Å resolution (Fig. 5c; Supplementary Fig. 6a, v-2, b, c). We docked models of FBF-2/ LST-1/RNA complexes in the final 3D reconstructions (Fig. 5b, c; right). However, the resolution was limited and we did not fit the models further. The 35-residue linker between the LST-1 PIMs as well as the three nucleotides between FBEa and FBEa* are expected to introduce structural flexibility in the relative arrangements of the two FBF-2 molecules. Nonetheless, the cryo-EM reconstruction provides a visual picture of how FBF-2, LST-1^19–98, and FBEa-FBEa* RNA can form a quaternary complex with two FBF-2 molecules bound to LST-1 and a single RNA.

With the partial molecular model of the FBF-2/LST-1/RNA complex, two LST-1 orientations were possible in the quaternary complex (Supplementary Fig. 7a). In option 1, LST-1 PIM-B interacts with the FBF-2 RBD bound to FBEa, and PIM-A interacts with the RBD bound to FBEa*. Option 2 is the reverse. We designed FRET (Fluorescent Resonance Energy Transfer) experiments to distinguish between these two possibilities (Supplementary Fig. 7a). We introduced cysteine residues in LST-1^19–98 in place of S26 and S74 at positions five residues upstream of PIM-A and PIM-B, respectively. We then labeled C26 or C74 with Cy5 fluorescent dye via cysteine-maleimide conjugation. Finally, we formed complexes composed of Cy5-labeled LST-1, unlabeled FBF-2 RBD, and 5′-Cy3 labeled *gld-1* FBEa-FBEa* RNA, and measured FRET efficiency from Cy3 at 564 nm in the RNA to Cy5 at 668 nm in LST-1. For option 1, LST-1 PIM-B would be near the 5′ Cy3 end of the RNA, and therefore, C74-labeled LST-1^19–98 would produce a higher FRET signal than C26-labeled LST-1^19–98. For option 2, LST-1 PIM-A would be near the 5′ Cy3 end of the RNA, and C26-labeled LST-1^19–98 would produce a higher FRET signal than C74-labeled LST-1^19–98. We measured FRET efficiency for pre-assembled complexes with a constant 2 μM FBF-2 RBD and varied ratios of LST-1:*gld-1* RNA. Under all conditions tested,

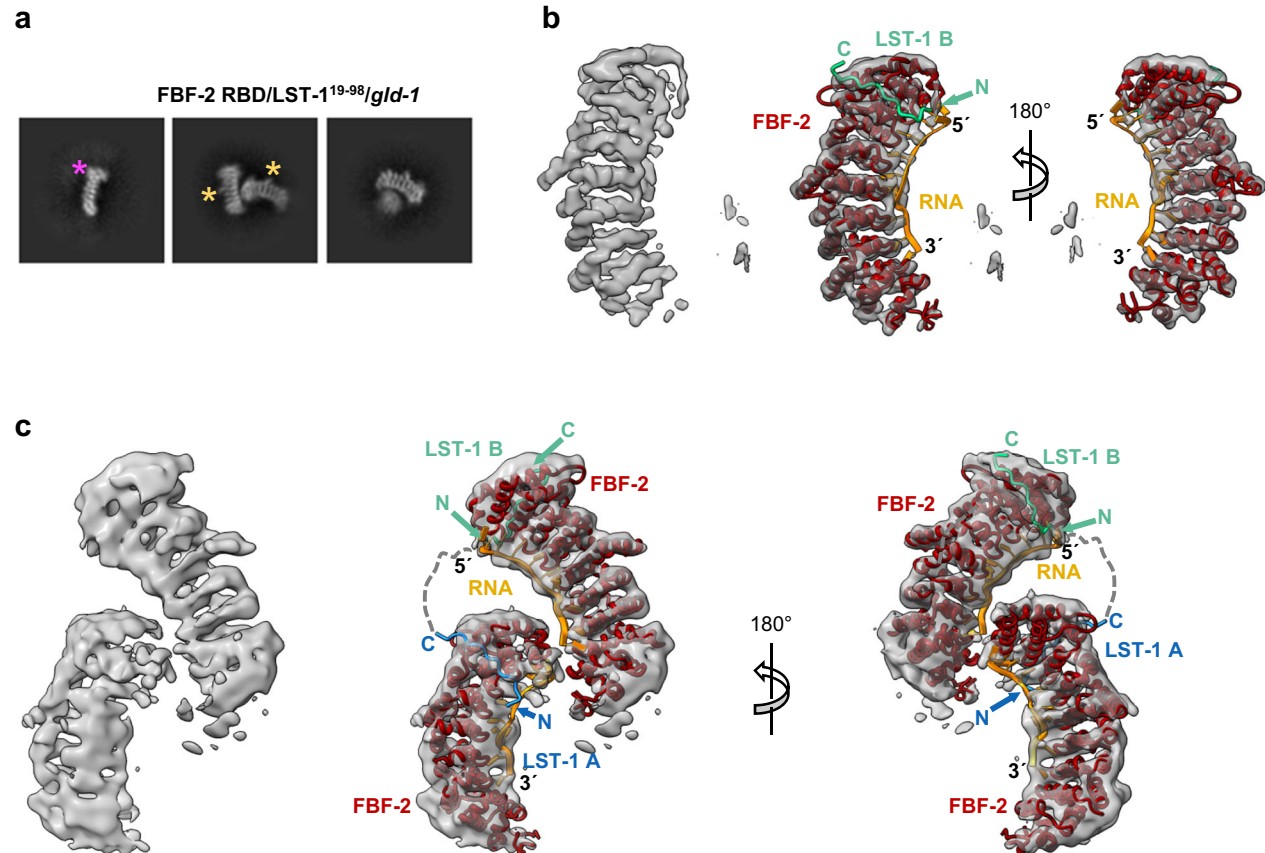

**Fig. 5 | Cryo-EM models of FBF-2/LST-1/*gld-1* RNA complex. a** Representative 2D classes of FBF-2 RBD/LST-1[19–98]/*gld-1* FBEa-FBEa*. The magenta asterisk marks a wider C-terminal end of the FBF-2 RBD due to longer helices in PUF repeat 8, and yellow asterisks mark a bump due to a long helix in the central repeat 5. See Supplementary Fig. 5c for all 2D classes. **b** Final 3D reconstruction of a single FBF-2 molecule (left) with superimposed molecular model of FBF-2 RBD/LST-1 B[76–90]/*gld-1* FBEa in two views (right). The model is derived from the aligned crystal structures of FBF-2 RBD/*gld-1* FBEa (PDB ID: 3V74) and LST-1[76–90] peptide (PDB ID: 6PUN). FBF-2 is shown in red, LST-1 B[76–90] in green, and FBEa in orange. **c** Final 3D reconstruction of FBF-2 RBD/LST-1[19–98]/*gld-1* FBEa-FBEa* RNA (left) with superimposed molecular models of FBF-2 RBD/LST-1[76–90]/*gld-1* FBEa and FBF-2 RBD/LST-1[27–41] (PDB ID: 7SO2) and *gld-1* FBEa* (PDB ID: 8VIV) in two views (right). LST-1 A[27–41] is shown in blue. The disordered linker between the LST-1 PIM A C-terminus (residue 41) and LST-1 PIM B N-terminus (residue 76) is indicated with a dashed line.

C74-labeled LST-1[19–98] yielded higher FRET efficiency than C26-labeled LST-1[19–98] (Supplementary Fig. 7b, columns 1 and 2). We conclude that option 1 with LST-1 PIM-B interacting with the FBF-2 RBD bound to FBEa is the predominant complex, although option 2 may also form with lower frequency.

**Adjacent binding elements in other FBF target mRNAs**

To ask whether closely spaced or adjacent FBEs exist in other mRNAs regulated by FBF-2, we examined eCLIP datasets[31] for RNAs with peaks over adjacent sites. Among 2,702 FBF-2 peaks in the dataset, we identified peaks covering one 9-nt canonical FBE (5′-UGURnnAUn-3′, where R = A/G and n=any nucleotide), like *gld-1* FBEa, plus a second element within 20 nts upstream or downstream (Fig. 6a). The second element could be another canonical FBE, a cFBE (5′-UGURnAUn-3′) or FBEa*-like (5′-UGUYRnAUn-3′ or 5′-UGUnRnnAUn-3′, where Y = C/U). This analysis identified 115 peaks with adjacent FBEs (113 in 3′UTRs and 2 in 5′UTRs) of 105 different RNAs, including the *gld-1* FBEa-FBEa* pair (Fig. 6b, c, Supplementary Data 1). This is a conservative estimate of the number of RNAs controlled by multiple FBEs, because we required one canonical FBE and that the second FBE be within 20 nucleotides. Thus, FBF-2 may regulate many RNAs via adjacent binding sites.

Next, we calculated the average height of peaks with adjacent FBEs. These peak heights were significantly higher (2825 fragments per kilobase mapped, FPKM) than all FBF-2 peaks (736 FPKM, Fig. 6d). This is consistent with data showing FBF-2 binding synergy to *gld-1* FBEa-FBEa* RNA (Fig. 4, Supplementary Fig. 4). We also examined whether

binding to adjacent sites was dependent on FBF-2 Y479, the residue critical for interaction with PIMs of various partner proteins and the FBF-2 PIM-CT. FBF-2 Y479A disrupts those interactions[9,27,31,41,42]; therefore, RNAs where FBF-2 occupancy changes with Y479A are likely regulated by higher order complexes that include PIM-dependent PUF partners and/or the FBF-2 PIM-CT. Among the 115 peaks with adjacent sites, peaks that were diminished upon loss of protein partnerships (Y479A) had the highest occupancy in wild-type animals (Fig. 6e). Taken together, these findings are consistent with our data that LST-1 (and potentially other partners) increase affinity of a higher order complex at adjacent sites.

Finally, we asked if RNA targets with adjacent sites under a single peak had related biological or molecular functions. Using gene ontology (GO) analysis[43], peaks with adjacent sites were enriched for some terms that are also enriched among FBF-2 targets more generally (Supplementary Fig. 8a–c)[21,44]. Enriched terms included regulation of translation and the cell cycle for biological processes and synaptonemal complex and P-granule proteins for cellular component. Taken together, we suggest that FBF-2 may regulate biological functions via higher order complexes on adjacent FBEs in many FBF-2 targets, in addition to *gld-1*.

## Discussion

Two discoveries reported here advance our understanding of how PUF proteins and their partners regulate RNAs and stem cells. The first is a ternary protein complex where two FBF-2 molecules bind to each of

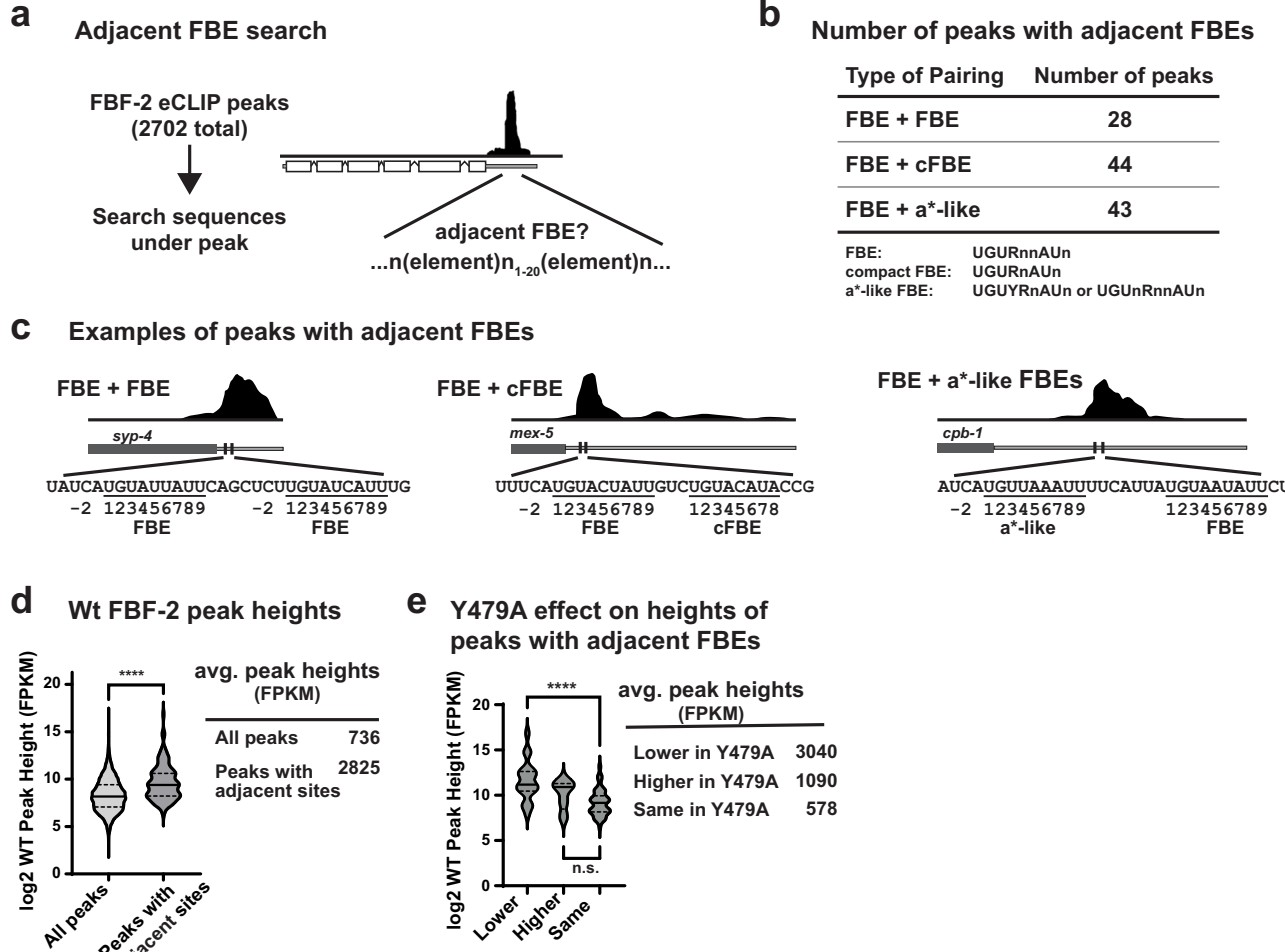

**Fig. 6 | Adjacent FBEs in >100 FBF-2 targets. a** Strategy to search for adjacent FBEs. Sequences under FBF-2 eCLIP peaks[31] were searched for one canonical FBE (called FBE) plus one other (canonical, compact, or FBEa*-like) with spacing between FBEs limited to 1-20 nucleotides. **b** >100 adjacent FBEs found. FBE definitions shown below. R = A/G, Y = C/U, n=any nucleotide. **c** Examples of peaks with adjacent FBEs. Dark gray rectangle, coding region; gray line, 3'UTR; FBEs, vertical black lines. Above, FBF-2 eCLIP peaks; below, FBE nucleotides numbered according to convention. **d** Violin plot of wild-type (wt) FBF-2 peak height of all peaks compared to those peaks with adjacent FBEs (FPKM = fragments per kilobase mapped). Solid horizontal line, mean height; dotted lines, quartiles. **** p = 1.4 × 10⁻¹⁶. **e** Peaks with adjacent sites categorized according to FBF-2 Y479 dependence: peak height lower in Y479A than wild-type, peak height higher in Y479A than wild-type, or peak height same (unchanged) in Y479A and wild-type. **** p = 1.2×10⁻⁸, n.s. = not significant (p = 0.11). P values calculated using a two-sided unpaired t-test assuming equal variance, no corrections for multiple comparisons.

---

two interaction motifs in one LST-1 molecule. The second is a pair of adjacent RNA regulatory elements in *gld-1*, FBEa-FBEa*, that is bound by the LST-1–FBF-2 ternary complex in vitro and that facilitates two PUF regulatory modes, repression and activation, in vivo. Together, these findings lead to models of PUF function that expand their regulatory repertoire in nematode stem cells (Fig. 7a, b). They also suggest potential for parallels in other organisms (Fig. 7c), where small intrinsically-disordered proteins of unknown function abound and noncanonical and adjacent binding elements remain largely unexplored.

### Self-renewal hub ternary complexes expand regulatory potential

Discovery of the 1 LST-1 – 2 FBF-2 ternary complex has major implications for the nematode self-renewal hub of PUF and partner proteins, which works within a regulatory network to co-ordinate self-renewal and differentiation of GSCs[14]. That network includes Notch signaling from the niche, which activates the self-renewal hub, and downstream differentiation hubs (e.g. GLD-1)[45]. FBF-2 and LST-1 are

key components of the self-renewal hub[24,28]; the two LST-1 PIMs are both required for GSC self-renewal and also required for RNA repression[10,11]. An LST-1–FBF-2 partnership with 1:1 stoichiometry was previously suggested to drive self-renewal[10], an idea confirmed by finding that LST-1 mutants with a single PIM can maintain GSCs[11]. However, the in vitro requirement of both LST-1 PIMs for ternary complex formation and binding to adjacent FBEs, together with the in vivo importance of adjacent FBEs for RNA regulation, argue that the LST-1–FBF-2 ternary complex is crucial for full RNA repression and stem cell control.

Might LST-1 link other PUF proteins in the self-renewal hub? Four PUF proteins (FBF-1, FBF-2, PUF-3, and PUF-11) function in that hub, and LST-1 binds to all four[12,14]. Therefore, it seems likely that LST-1 forms ternary complexes with all four self-renewal PUFs and further that LST-1 may link different PUF proteins (e.g. FBF-1 and FBF-2 in one ternary complex). Although the four PUF proteins maintain GSCs with extensive biological redundancy, each has distinct features, likely due to singular aspects of regulation and biochemistry (e.g[14,46,47].). For example, FBF-1 and FBF-2 bind to RNA with comparable sequence

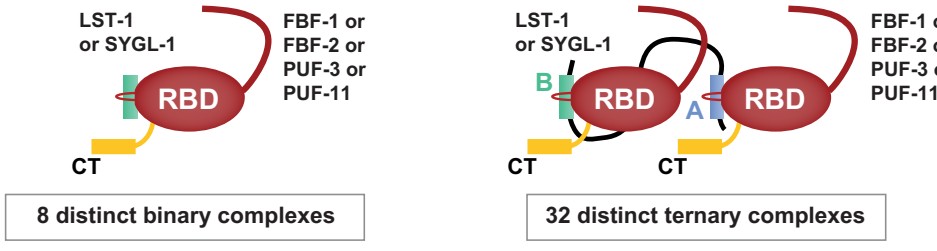

**a** Expansion of hub repertoire

8 distinct binary complexes

32 distinct ternary complexes

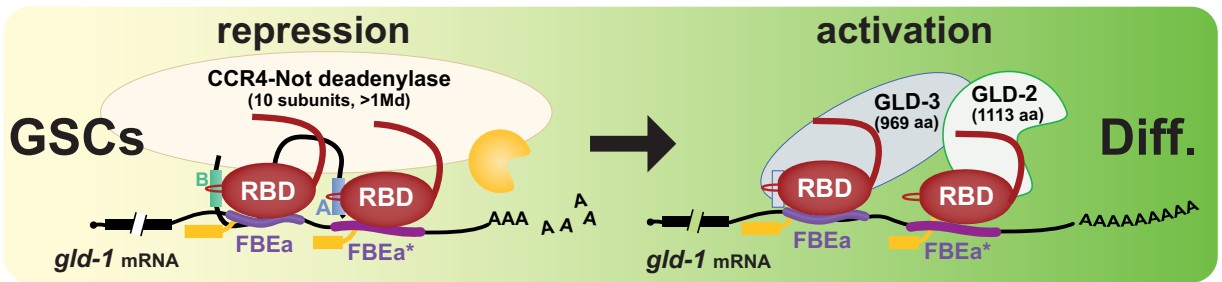

**b** Model: Higher order complexes enhance *gld-1* RNA regulation in nematode distal germline

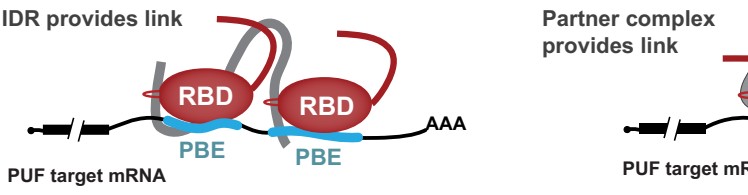

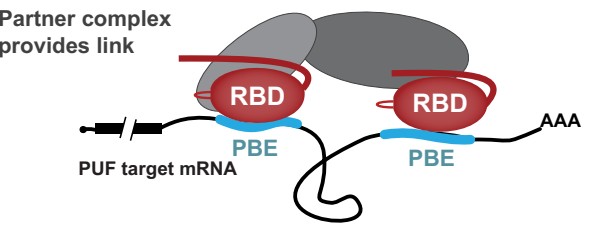

**c** Models: IDRs and effector complexes have potential to link two PUF proteins more broadly

**Fig. 7 | Expanding the FBF regulatory repertoire. a** Ternary complexes likely expand regulatory repertoire of the self-renewal hub. Eight partnerships with one of four self-renewal PUF proteins and either LST-1 or SYGL-1 were proposed previously (left)[14]. This work posits additional ternary complexes (right), each with any two of the four self-renewal PUF proteins plus either LST-1 or SYGL-1. Binding to adjacent FBEs expands the repertoire further, because each PUF protein can bind to either of the two FBEs (not shown). **b** Models: Higher order complexes enhance RNA regulation in vivo. Models for RNA repression in GSCs (yellow) and for RNA activation in differentiating germ cells (green). Left, stronger RNA-binding affinity of the FBF-2–LST-1 ternary complex and its binding to two FBEs increases likelihood of recruiting CCR4-NOT deadenylase (see Discussion). Right, GLD-2/GLD-3 poly(A) polymerase recruited to mRNA by established interactions: GLD-3 PIM (light blue rectangle) with FBF-2[42,55] plus GLD-2 interaction with FBF-2[25]. Synergistic binding of FBF-2 to adjacent FBEs increases likelihood of poly(A) polymerase recruitment and activation of translation. **c** Models for generality of higher order PUF protein complex, linked either by an LST-1-like mechanism (left) or a GLD-2/GLD-3-like mechanism (right).

specificity[20,23], which is related to but distinct from PUF-11 RNA-binding specificity[48]. In addition to the possibility of LST-1 linking two different PUF proteins, another PUF partner in the self-renewal hub, SYGL-1, may also link PUF proteins to create ternary complexes. Not only are LST-1 and SYGL-1 redundant for GSC self-renewal[28], but like LST-1, SYGL-1 possesses two PIMs within an extended IDR and binds to all four self-renewal PUFs[10,14]. The four self-renewal PUF proteins and two partners were previously suggested to form eight partnerships that drive self-renewal[14]. However, the potential for LST-1 and SYGL-1 to form ternary complexes with any two of the four PUFs would tremendously expand the regulatory capability of the self-renewal hub (Fig. 7a). Such an expansion would strengthen the self-renewal hub and allow it to maintain GSCs in the face of diverse developmental, physiological, and environmental challenges.

## FBEa-FBEa* adjacent sites in *gld-1* 3′UTR are part of a higher order complex

From discovery of the LST-1–FBF-2 ternary complex, we hypothesized that its two PUF proteins might bind to adjacent FBEs in target RNAs. This idea led to identification of FBEa-FBEa*, a candidate FBE pair in the *gld-1* 3′UTR (Fig. 2). FBEa is a canonical element and well established[20,23,24,33], but its FBEa* neighbor has a divergent sequence that was not recognized as an FBE previously. Occupancy of FBF-2 on the *gld-1* 3′UTR region harboring FBEa-FBEa* was exceptionally prominent in vivo, based on CLIP data[21,31]. We do not know the order of assembly of FBF-2 and LST-1 on the adjacent *gld-1* FBEs, but suggest that the higher affinity FBEa RNA element and stronger LST-1 PIM-B increase effective protein concentrations to promote engagement of FBF-2 with the lower affinity FBEa* and weaker LST-1 PIM-A.

Many FBF-2 target mRNAs have two sites under a single peak of FBF-2 binding (Supplementary Data 1). One example is *syp-3* mRNA, which encodes a meiotic synaptonemal complex protein. Adjacent *syp-3* FBEs are required for repression[33], although individual FBE contributions were not tested. Y479A, a mutant FBF-2 that does not bind LST-1, has lower occupancy of *syp-3* adjacent FBEs than wild-type[31], consistent with regulation by the LST-1–FBF-2 ternary complex. A second example is *fog-1* mRNA, which encodes a sperm fate regulator. Adjacent FBEs in the *fog-1* 3'UTR promote strong FBF-2 binding and an FBF-2 supershift in vitro, suggesting interaction with more than one FBF-2 molecule[49]. Although occupancy was unchanged in Y479A[31], FBF represses *fog-1* in larvae and occupancy was assayed in adults. Similar differences in developmental state may explain why FBF-2 Y479A occupancy did not decrease at other adjacent sites identified in the eCLIP dataset. Regardless, the many adjacent FBEs in the *C. elegans* transcriptome broaden the potential for ternary complex regulation of RNA expression.

### FBEa FBEa* enhances both RNA repression and activation

The FBEa FBEa* adjacent sites affect two regulatory modes of FBE-mediated *gld-1* regulation. One mode is *gld-1* repression, which occurs in GSCs where both FBF-2 and LST-1 are present (Fig. 7b, left). *gld-1* repression in this region is normal in FBEb single mutants, which leave FBEa FBEa* intact, but compromised in both FBEa mutants and FBEa* single mutants and the FBEa FBEa* double mutant. A simple explanation is that FBEa or FBEa* alone cannot mediate full repression of GLD-1 and suggests that binding of the LST-1–FBF-2 ternary complex to FBEa FBEa* RNA elements is critical for full repression. Repression depends on recruitment of the CCR4-NOT (CNOT) deadenylase complex (Fig. 7b, left), and full repression likely depends on enhanced recruitment and deadenylase activity. The enzymatic CCF-1 subunit of CNOT interacts directly with FBF-2 and promotes *gld-1* repression in the distal germline[25], and the scaffold NTL-1/NOT-1 subunit immunoprecipitates with both FBF-2[31] and LST-1[10]. An LST-1 mutant lacking its PIMs, however, does not immunoprecipitate with NTL-1/NOT-1, suggesting that LST-1 associates indirectly with the scaffold subunit via PUF proteins[10]. We also posit that the two FBF-2 N-terminal IDRs in the FBF-2–LST-1 ternary complex increase the probability of CNOT recruitment. This idea is based on PUF N-terminal IDRs recruiting CNOT in yeast, flies and mammals, and longer IDRs making multivalent interactions, recruiting CNOT more effectively than shorter IDRs[50–53]. Our current model is that the LST-1–FBF-2–FBEa-FBEa* higher order complex increases likelihood of forming a repressive complex with CNOT deadenylase.

The second regulatory mode is RNA activation. This occurs in differentiating germ cells where FBF-2 remains but LST-1 is no longer present (Fig. 7b, right). Activation of *gld-1* RNA occurs normally in single mutants for any of the three elements (FBEa, FBEa*, and FBEb), but it is compromised upon loss of any two. Therefore, *gld-1* activation requires two FBEs, but the pair can either be adjacent as in FBEa FBEa* or separated as in the FBEa FBEb and FBEa* FBEb pairs. Although separated FBEs exist in other mRNAs (e.g[54],), prior studies have not explored their individual functions or synergy. How two FBEs enhance *gld-1* RNA activation is not understood, but previous studies provide a possible mechanism. Two other FBF partners, GLD-2 and GLD-3, are expressed in differentiating germ cells[55,56], and genetically FBF works in the GLD-2/GLD-3 pathway to promote meiotic entry[24]. GLD-2 and GLD-3 are subunits of a heterodimeric poly(A) polymerase that promotes meiotic entry[36,56–58]. FBF interacts directly with GLD-2 and stimulates GLD-2 polyadenylation of an FBE-containing RNA in vitro[25]; FBF also interacts directly with GLD-3 via a PIM in the C-terminal region of GLD-3 (Fig. 7b, right, blue rectangle)[42,55]. Putting these pieces together, we suggest that two FBEs can anchor a quaternary complex of GLD-2, GLD-3, and two PUF proteins (Fig. 7b, right). The GLD-2–GLD-3 heterodimer is much larger than the LST-1 IDR, and therefore may accommodate binding two FBF-2 molecules at either adjacent or separated FBEs. As

noted for the CCR4-NOT deadenylase, our model hypothesizes that multiple FBEs and higher affinity FBEs increase the likelihood of recruiting GLD-2/GLD-3 for RNA activation.

### Potential for broad use of higher order PUF complexes

Our discoveries of the LST-1–FBF-2 ternary complex and adjacent RNA regulatory elements take our understanding of nematode stem cell regulation to a new level. Collaborations between PUF proteins and partners can be impacted by strength and number of PIMs; strength, number, and spacing of FBEs; spatiotemporal distribution of PUF partners; and engagement of different effector enzymes. Together, these attributes have potential for tremendous diversity of regulatory outcomes with profound impacts on the fate of many mRNAs. This previously unrecognized diversity may well be fundamental to plasticity and evolution of RNA regulatory networks. The concept that PUF proteins and their partners can form higher order complexes that enhance regulatory complexes via multiple FBEs opens the door to investigating the existence, regulation, and function of additional higher order PUF complexes in nematodes and more broadly throughout the phylogenetic tree, including humans. Though not yet found outside nematodes, we envision that higher order PUF complexes may operate broadly among Eukarya and that they regulate RNAs via either adjacent PUF binding elements (Fig. 7c, left) or more distant PUF binding elements (Fig. 7c, right). The current lack of other examples likely reflects the prevalent focus on proteins with structural domains and on sequence elements that match the expected consensus and have optimal RNA-binding affinity. Yet we have found that the intrinsically-disordered region of LST-1 is responsible for assembling a functionally significant ternary complex and that the weaker binding noncanonical FBEa* element is critical for patterned RNA regulation and fertility. We speculate that the LST-1–FBF-2 ternary complex and its binding to adjacent FBEs may be emissaries of a broadly used regulatory strategy of major significance.

## Methods

### Protein expression and purification

*C. elegans* FBF-2 RBD+CT (residues 164-632) was expressed and purified as described previously[9]. The FBF-2 RBD+CT protein was overexpressed in *E. coli* BL21-CodonPlus (DE3)-RIL competent cells (Agilent). A 1-L LB culture was inoculated with a 5-mL overnight culture and grown at 37 °C to $OD_{600}$ of ~0.6. Protein expression was induced with 0.1 mM IPTG, and the culture was grown at 16 °C overnight. The $His_6$-SUMO-tagged FBF-2 RBD+CT protein was purified from the soluble fraction of *E. coli* cell lysate in a buffer containing 20 mM Tris pH 8.0, 0.5 M NaCl, 20 mM imidazole, 5% (v/v) glycerol, and 0.1% (v/v) β-mercaptoethanol using Ni-NTA resin (Qiagen), and was eluted with a buffer containing 20 mM Tris pH 8.0, 50 mM NaCl, 0.2 M imidazole, and 1 mM DTT. The eluted fusion protein was incubated with Ulp1 protease overnight to remove the $His_6$-SUMO tag. Subsequently, the FBF-2 protein was purified with a Hi-Trap Heparin column (Cytiva), eluting with a 5-100% gradient of buffer B. Heparin column buffer A contained 20 mM Tris pH 8 and 1 mM DTT, and buffer B contained an additional 1 M NaCl. The peak fractions from the heparin column were pooled and concentrated to 5 ml and loaded onto a HiLoad 16/60 Superdex 75 column (Cytiva) in a buffer containing 20 mM HEPES pH 7.4, 150 mM NaCl, and 0.5 mM TCEP. Purified FBF-2 RBD+CT protein was concentrated and snap frozen for later binding experiments.

The FBF-2 RBD protein (residue 164-575) was expressed and purified as described previously[59]. TEV protease was used to cleave the His-SUMO tag because it was generated from a previous pGEX6p FBF-2 RBD construct. Otherwise, the purification procedure was the same as the FBF-2 RBD+CT protein.

A codon-optimized synthetic cDNA fragment encoding LST-1 residues 19-98 was cloned into the pSMT3 vector. The LST-1[19–98] protein was expressed in *E. coli* BL21-CodonPlus (DE3)-RIL cells (Agilent).

A 1-L TB culture was inoculated with a 5-mL overnight culture and grown at 37 °C to $OD_{600}$ of ~0.6. Protein expression was induced with 0.4 mM IPTG, and the culture was grown at 22 °C overnight. The LST-$1^{19-98}$ protein was purified from the soluble fraction of *E. coli* cell lysate using Ni-NTA resin (Qiagen). The eluted fusion protein was incubated with Ulp1 protease overnight to remove the $His_6$-SUMO tag. Subsequently, the LST-$1^{19-98}$ protein was purified with a Hi-Trap Heparin column (Cytiva), eluting with a 5–100% gradient of buffer B. The Ni-NTA and heparin column buffers were the same as those used for FBF-2 RBD+CT protein purification. The peak fractions from the heparin column were pooled and passed through a Ni-NTA gravity column (Qiagen) to remove released $His_6$-SUMO protein. The LST-$1^{19-98}$ protein was finally loaded onto a HiLoad 16/60 Superdex 75 column (Cytiva) in a buffer containing 20 mM HEPES pH 7.4, 150 mM NaCl, and 0.5 mM TCEP. Purified LST-$1^{19-98}$ protein was concentrated and snap frozen for later binding experiments.

LST-$1^{19-98}$ variants were generated by mutagenesis PCR, and the nucleotide sequences were confirmed by Sanger sequencing: LST-$1^{19-98}(A^m)$: K32A, L35A; LST-$1^{19-98}(B^m)$: K80A, L83A, Y85A; LST-$1^{19-98}(A^m B^m)$: K32A, L35A, K80A, L83A, Y85A. Mutant LST-1 proteins were prepared using the same protocol as for WT protein.

LST-1 peptides carrying PIM A (residues 19-50) or PIM B (residues 67-98) were purified as described previously[27]. *E. coli* BL21-CodonPlus (DE3)-RIL competent cells (Agilent) were transformed with cDNAs encoding LST-$1^{19-50}$ or LST-$1^{67-98}$ in the pSMT3 vector. A 5-mL culture was grown overnight at 37 °C and then used to inoculate 1 L of TB media with 50 μg/mL kanamycin. The culture was grown at 37 °C. Protein expression was induced at $OD_{600}$ of ~1.0 with 0.4 mM IPTG, and the culture was grown at 22 °C for ~20 h. The soluble fraction of *E. coli* cell lysate in a buffer containing 20 mM Tris, pH 8.0; 0.5 M NaCl; 20 mM imidazole; and 5% (v/v) glycerol was mixed with 5 mL Ni-NTA resin (Qiagen) for 1 h at 4 °C. After extensive washing the LST-1 proteins were eluted with a buffer of 20 mM Tris, 50 mM NaCl and 200 mM imidazole, pH 8. The Ulp1 protease was added to the eluant and incubated at 4 °C for 2 h to cleave the $His_6$-SUMO tag from LST-1. LST-$1^{19-50}$ protein was separated from $His_6$-SUMO with a HiTrap Q column (Cytiva) and the column flow-through containing LST-$1^{19-50}$ was collected and concentrated using Amicon Ultra-15 filters (3 K MWCO). LST-$1^{67-98}$ protein was purified with a HiTrap Heparin column (Cytiva) and eluted with a 5-100% NaCl gradient (buffer A: 20 mM Tris, pH 8.0; buffer B: 20 mM Tris, pH 8.0, 1 M NaCl). The peak fractions containing LST-$1^{67-98}$ were concentrated. Both LST-$1^{19-50}$ and LST-$1^{67-98}$ were further purified with a HiLoad 16/60 Superdex 75 column (Cytiva).

## Isothermal Titration Calorimetry

Experiments were performed at 20 °C using a MicroCal PEAQ-ITC Automated (Malvern Instruments) with a 200-μL standard cell and a 40-μL titration syringe. FBF-2 and LST-1 variants were prepared in a buffer of 20 mM HEPES pH 7.4, 150 mM NaCl, and 0.5 mM TCEP by gel filtration. All protein concentrations were determined by NanoDrop based on UV absorption at 280 nm. LST-1 (100-400 μM) was titrated from the syringe into the cell containing FBF-2 (10-30 μM) in 2 μl aliquots. Experiments were performed in duplicate due to the limitation of the amount of protein needed. Data were analyzed with the one-site model using the MicroCal PEAQ-ITC analysis software provided by the manufacturer. We also integrated the data with NITPIC[60] and attempted to fit with a two-site model using the SEDPHAT program[61], but without success. Fitting a model for two independent binding events to a uniphasic curve is challenging due to the need to refine multiple thermodynamic parameters without constraints[62].

## Nematode strain maintenance

*C. elegans* were maintained using standard culture conditions[63]. Wild-type *C. elegans* and strains carrying mutations in the endogenous *gld-1* locus were maintained at 20 °C. GFP reporter strains were maintained at 24 °C.

## CRISPR/Cas9 genome editing to create FBE mutations

Mutations in FBEa and FBEa* sites in the *gld-1* 3′UTR were created by co-CRISPR editing using a CRISPR/Cas9 RNA-protein complex[64,65]. Wild-type or reporter animals carrying a wild-type *gld-1* 3′UTR[66] were injected with a mix containing a gene-specific crRNA (5 μM, IDT-Alt-R™), unc-58 repair oligo (1 μM, IDT), tracRNA (4.5 μM, IDT), gene-specific repair oligo (5 μM, IDT) and Cas9 protein (glycerol free, 3 μM, IDT). F1 progeny of injected hermaphrodites were screened for edits by PCR using one primer specific for either endogenous *gld-1* or the GFP reporter and another specific for the FBE being edited, sequenced, and outcrossed against wild-type strain prior to analysis. Sequences of guide RNAs and repair templates are in Supplementary Table 5, and strains used in this manuscript as well as primers used for screening are in Supplementary Table 6.

## Immunohistochemistry

For endogenous GLD-1, we used a strain containing GFP in all somatic nuclei (*sur-5::GFP*) as our wild-type control. This allowed us to use GFP to identify wild-type animals so that we could process wild-type and FBE mutants together. Worms were grown at 20 °C. Wild-type and FBE mutant hermaphrodites were picked as mid-L4s, and gonads were dissected together 22–26 h later (A24 adults) and then processed in the same tube. For GFP reporter strains, we compared GFP levels in a strain with a wild-type 3′UTR to strains with FBE mutations in the 3′UTR. To reduce variability due to GFP silencing, GFP reporters were grown at 24 °C and were outcrossed with *oma-1::GFP*, a strain that licenses GFP expression[67]. Wild-type and FBE mutant reporter hermaphrodites were picked as mid-L4, and gonads from each genotype were dissected separately 20-24 h later and processed in separate tubes.

Gonads were extruded, fixed, and stained using standard protocols[68]. Animals were removed from plates in ~200 μl PBS containing 0.1% (v/v) Tween 20 (PBSTw) and 0.25 mM levamisole and pipetted into a multiwell dish. They were then cut behind the pharynx to extrude the gonad and pipetted into a 1.5 ml microfuge tube and 4% (v/v) paraformaldehyde was added for 10 min at room temperature. After centrifugation at 160 x g for 30 s, excess liquid was removed, and gonads were permeabilized with 100 μl PBSTw containing 0.5% (w/v) BSA (PBSBTw) for 5 min at room temperature. Centrifugation was repeated, and gonads were blocked in PBSBTw for 30 min at room temperature (for anti-GLD-1) and PBSBTw containing 30% goat serum for anti-GFP. Primary antibodies were added, and tubes were placed at 4 °C overnight. Rabbit anti-GLD-1[69] and mouse anti-GFP (3E6, ThermoFisher Scientific, Catalog # A-11120 RRID:AB_221568) were each used at 1:200 dilution. Secondary antibodies (Alexa 488, ThermoFisher Scientific Catalog #A21202 RRID:AB_141607; Alexa 647, Jackson ImmunoResearch Catalog # 711-605-152 RRID:AB_2492288) were used at 1:1000 dilution in PBSBTw and incubated for 1 h at room temperature on a rotating rack shielded from light, followed by 3 washes in PBSBTw. Finally, samples were mounted in 10 μl ProLong Gold (ThermoFisher) and covered with a 22x22 coverslip, allowed to cure overnight to several days, and imaged on a Leica SP8 confocal microscope using HyD detectors for GLD-1 and GFP, PMT detectors for DAPI. Endogenous GLD-1 was imaged at 63x magnification, and GFP was imaged at 40x magnification.

## Quantitation and analysis of GLD-1 and GFP protein levels

GLD-1 and GFP protein levels were quantitated in summed projections of confocal stacks using ImageJ as described in ref. 31. Briefly, a line (linewidth = 50 pixels for GLD-1, 30 pixels for GFP) was drawn along the distal-proximal axis of the germline. Pixel intensity was measured

using the plot profile function. Intensities were copied into Excel. Data were then copied into Graphpad Prism (v. 9.4.1-10.2.2), and the Row Statistics function was used to calculate mean values along the distal-proximal axis. We normalized the means of both wild-type and FBE mutants to the maximum level of GLD-1 or GFP found in the distal 100 microns of wild-type germlines[32]. The normalized means were then graphed with 95% confidence intervals. Representative images are z-projections of confocal stacks.

For Fig. 3h, we assigned GLD-1 levels "+" symbols as indicated in footnote. For the most part the levels were clear and consistent between endogenous GLD-1 and reporter. Two cases that were less clear are described here. Line 3: The small increase in endogenous GLD-1 and reporter GFP levels in FBEb$^m$ distal 0-10 μm was not significant for endogenous GLD-1 but was significant for reporter GFP[31]. The endogenous levels are significantly lower than GLD-1 levels in the ++ category. Thus we assigned it "+". Line 7: The decrease in endogenous and reporter FBEa$^m$ FBEb$^m$ in the proximal 90-100 μm was not significant for endogenous GLD-1 but was significant for reporter GFP[31]. The decrease in GLD-1 levels in endogenous FBEa$^m$ FBEb$^m$ in the proximal 90–100 μm was significantly different from that seen in FBEa'$^m$ FBEb$^m$ and FBEa$^m$ FBEa*$^m$ in the proximal 90–100 μm, thus we assigned it "++++".

## Phenotypic analysis

Adult animals were scored as fertile (visible oocytes and embryos in gonad) or sterile (no oocytes or embryos in gonad) using a dissecting microscope. Sterile animals were then either mounted on agarose pads and scored for germ cell phenotypes on a compound microscope or stained with DAPI and scored based on nuclear morphology using fluorescence microscopy GSCs are undifferentiated and both their cellular and nuclear morphologies are distinct from differentiated gametes. Sperm and oocytes also have distinctive morphologies. A gonad was scored as having GSCs if undifferentiated germ cells were adjacent to the distal tip cell in the distal gonad, and as lacking GSCs if sperm were adjacent to the distal tip cell in the distal gonad. A gonad was scored as positive for the sperm-to-oocyte switch if both sperm and oocytes were visible in the gonad. If only sperm were present, the gonad arm was scored as negative for the sperm-to-oocyte switch. The data are presented as percent gonad arms that have GSCs and percent gonad arms that make the sperm to oocyte switch. All gonad arms that lack GSCs also fail to make the sperm-to-oocyte switch. The two individual gonadal arms in a single animal can have different GSC and s/o switch phenotypes.

## X-ray crystal structure determination

Purified FBF-2 RBD protein ($A_{280}$ = 3.05) was incubated with the *gld-1* FBEa* RNA (5'-CAUGUUGCCAU-3') at a molar ratio of 1:1.2 on ice for 1 h prior to crystallization. Crystals of the protein-RNA complex were grown in 10% (v/v) PEG 20000, 2% (v/v) dioxane and 0.1 M bicine pH 9.0 by hanging drop vapor diffusion at 20 °C with a 1:1 ratio of sample:reservoir solution. Crystals were cryoprotected by transferring crystals into crystallization solution supplemented with 5%, 10% and 20% (w/v) glycerol sequentially and then flash frozen in liquid nitrogen.

X-ray diffraction data were collected at a wavelength of 1.0 Å at beamline 22-ID of the Advanced Photon Source. Data sets were scaled with HKL2000[70]. The crystals belonged to the P6$_1$ space group. An asymmetric unit contained one binary complex. The structure of the FBF-2 RBD/*gld-1* FBEa binary complex (PDB code: 3V74) was used as a search model for molecular replacement with Phaser[71]. The model was improved through iterative refinement and manual building with Phenix[72] and Coot[73]. Data collection and refinement statistics are shown in Supplementary Table 1. 98% of residues were in the most favored regions of the Ramachandran plot and there were no outliers.

## EMSA RNA-binding assays

3'-Cy5-labeled synthetic RNAs were ordered from IDT (Supplementary Table 4). RNA (5 nM) was mixed with serially-diluted protein samples in a buffer of 10 mM HEPES pH 7.4, 75 mM NaCl, 0.01% (v/v) Tween 20, 0.1 mg/ml BSA, 0.1 mg/ml yeast tRNA, and 2 mM DTT. The FBF-2 proteins were diluted in 2-fold series from the highest concentration of 5 μM. For experiments in the presence of LST-1, LST-1 variant proteins at constant concentration were preincubated with FBF-2 at 4 °C for 2 h: 150 μM LST-1$^{19-50}$, 50 μM LST-1$^{67-98}$, and 10 μM LST-1$^{19-98}$. The concentrations were chosen based on their $K_D$'s for interaction with FBF-2. The protein-RNA mixtures were incubated at 4 °C overnight. The samples were resolved on 10% TBE polyacrylamide gels run at constant voltage (100 V) with 1x TBE buffer at room temperature for 35 min. The gels were scanned and visualized with a Typhoon FLA 9500 imager using the Cy5 channel (excitation wavelength 635 nm). Band intensities were quantitated with ImageQuant 5.2. The data were fit with GraphPad Prism using nonlinear regression with a model for specific binding with a Hill slope. For experiments with the FBEa-FBEa* RNA, fitting with a two-site binding model failed. Therefore, we used the model for specific binding with Hill slope and report apparent dissociation constants. Hill slope values for binding to either FBEa* or FBEa-FBEa* RNAs were all greater than 1. Mean $K_{D,app}$ and SEM from three or more technical replicates are reported (Table 2).

## Purification of FBF-2 RBD or RBD+CT/LST-1$^{19-98}$ complex

The FBF-2 RBD or RBD+CT/LST-1$^{19-98}$ complex was purified using the same protocol as described previously for the FBF-2 RBD/LST-1 B complex[27]. Briefly, the His$_6$-SUMO fused FBF-2 RBD or RBD+CT and the GST-fused LST-1$^{19-98}$ were coexpressed in *E. coli* BL21-CodonPlus (DE3)-RIL cells (Agilent). The soluble fraction of *E. coli* cell lysate in a buffer containing 20 mM Tris, pH 8.0; 0.5 M NaCl; 20 mM imidazole; 5% (v/v) glycerol; and 0.1% (v/v) β-mercaptoethanol was mixed with 5 mL Ni-NTA resin (Qiagen) for 1 h at 4 °C. After washing the beads, the His$_6$-SUMO-FBF-2 and GST-LST-1$^{19-98}$ fusion proteins were co-eluted with elution buffer (20 mM Tris, pH 8.0; 50 mM NaCl; 200 mM imidazole, pH 8.0; 1 mM DTT). For the FBF-2 RBD/LST-1$^{19-98}$ complex, TEV protease was added to the eluent and incubated at 4 °C overnight to cleave the His$_6$-SUMO fusion from FBF-2 RBD and the GST fusion from LST-1$^{19-98}$. For the FBF-2 RBD+CT/LST-1$^{19-98}$ complex, both Ulp1 and TEV proteases were used to cleave the tags off. The FBF-2/LST-1$^{19-98}$ protein complex was purified with a Hi-Trap Heparin column (Cytiva), eluting with a 5–100% gradient of buffer B. Heparin column buffer A contained 20 mM Tris, pH 8.0 and 1 mM DTT, and buffer B contained an additional 1 M NaCl. The peak fractions were concentrated, and the protein complex was purified using a HiLoad 16/600 Superdex 200 column (Cytiva) in a buffer of 20 mM HEPES, pH 7.4, 0.15 M NaCl and 2 mM DTT. The purified FBF-2/LST-1 ternary complex was incubated with a 29-nt *gld-1* FBEa-FBEa* RNA at a molar ratio of 1:1.2 at 4 °C for 1 hour, and the quaternary complex was purified by a Superdex-200 10/300 GL column.

## SEC-MALS Analyses

The molecular masses of FBF-2 RBD+CT/LST-1$^{19-98}$/*gld-1* FBEa-FBEa* and FBF-2 RBD/LST-1$^{19-98}$/*gld-1* FBEa-FBEa* complexes were assessed by SEC-MALS using an AKTAPure chromatography system (Cytiva) coupled to miniDawn TREOS and Optilab rEX detectors (Wyatt Technology). 100 μl of purified FBF-2 RBD+CT or RBD/LST-1$^{19-98}$ protein (~30 μM) with the *gld-1* FBEa-FBEa* RNA (5'-AUCAUGUGCCAUACAU CAUGUUGCCAUUU-3') was run on a Superdex-200 10/300 GL column (Cytiva) in a buffer containing 20 mM HEPES pH 7.4, 150 mM NaCl, and 2 mM DTT. All data were analyzed using ASTRA 7.3.2.17 software (Wyatt Technology).

## Cryo-EM

Purified quaternary complexes of FBF-2 RBD/LST-1[19–98]/*gld-1* FBEa-FBEa* RNA ($A_{280}$ ~ 0.2, $A_{260}/A_{280}$ ~ 1.4, 3 µl) were deposited onto Ultra-AuFoil R1.2/1.3 300 gold mesh grids (QuantiFoil) that were plasma cleaned before use. The grids were back-blotted for 3 s, and vitrified using a Leica automatic plunge freezer.

Cryo-EM images were collected on a Talos Arctica electron microscope (Thermo Fisher Scientific) operated at 200 keV and equipped with a K2 Summit direct detection camera (Gatan). Movies were recorded in counting mode at a nominal magnification of 45,000x corresponding to 0.932 Å/pixel (Supplementary Table 3). CryoSparc was used for data processing, starting with patch motion correction and patch CTF estimation[40]. After an initial round of blob picking and 2D classification, Topaz was used for model training and model-based particle picking. Particles were extracted with a box size of 300 pixels and derived from areas of relatively low particle density. Particles were subjected to 2D classification. Ab initio reconstruction was used to generate initial models while varying the number of ab initio classes. Four 3D classes were generated (Supplementary Fig. 6a, iii). Upon inspection, one 3D class with two well-resolved FBF-2 molecules and one 3D class with one well-resolved FBF-2 molecule were selected, and the particles were input for a second round of ab initio reconstruction (Supplementary Fig. 6a, iv-1 and iv-2). The best subclass for 3D classes with one or two FBF-2 molecules was refined using non-uniform refinement (Supplementary Fig. 6a, v-1 and v-2) to 4.37 Å or 6.39 Å resolution, respectively. ChimeraX was used for volume viewing and fitting crystal structures[74]. The volume containing two FBF-2 molecules was flipped along the z axis in ChimeraX to get correct handedness to fit crystal structures. Cryo-EM analyses using FBF-2 RBD+CT instead of the RBD yielded similar 'monomer' and 'dimer' 2D classes, but greater heterogeneity of the particles due to flexibility limited the resolution of 3D reconstructions for the RBD+CT samples.

## FRET

Purified LST-1[19–98] S26C and S74C mutant proteins were each labeled using Cy5 maleimide dye. The *gld-1* FBEa-FBEa* RNA was synthesized with Cy3 at the 5′ end (5′-Cy3-AUCAUGUGCCAUACAUCAUGUUGCCAUUU-3′). 2 µM FBF-2 RBD protein was pre-incubated with Cy5-labeled LST-1[19–98] (20, 40, 80 nM) in a volume of 25 µl at room temperature for 30 min. 25 µl Cy3-labeled RNA was added to a final concentration of 20 nM (LST-1[19–98]:*gld-1* ratios were 0.5:1, 1:1, and 2:1) and incubated for 30 min. FRET measurements were carried out using a TECAN Infinite M1000 PRO plate reader. Cy3 was excited at 535 nm, and an emission scan was performed from 550 to 725 nm. Cy3 and Cy5 emission peaks were detected at ~564 nm and ~668 nm, respectively. Relative FRET efficiency was calculated as $I_{668}/(I_{668} + I_{564})$, where $I_{668}$ and $I_{564}$ are the measured fluorescence intensities. Samples of *gld-1* RNA only and RNA with LST-1[19–98] in the absence of FBF-2 were analyzed as negative controls.

## Bioinformatic analyses

Search for adjacent sites was performed with Biopython 1.81[75]. Briefly, previously published eCLIP peak sequences were converted to FASTA format and searched for exact matches to FBE variants using SeqIO[75]. Elements were defined as follows: 9-nt canonical FBE, 5′-UGURnnAUn-3′; cFBE, 5′-UGURnAUn-3′, and FBE a*like, 5′-UGUYRnAUn-3′ or 5′-UGUnRnnAUn-3′. Peaks with at least one FBE and one additional element were extracted regardless of 5′-3′ orientation (i.e. the second element could be upstream or downstream of the initial FBE). Distances between motifs were calculated using the starting position of each element, accounting for the length of the 5′-most element (start position of 3′-most element – start position of 5′-most element – length of 5′-most element = distance between motifs). Adjacent sites were limited to those with 1-20 nucleotides between motifs.

Gene ontology (GO) enrichment was performed with the database for annotation, visualization, and integrated discovery website (v. DAVID 2021)[43]. Significance of GO enrichment was calculated by FDR. GO terms were included if the term was >2 fold enriched over background and had an FDR ≤ 0.05.

## Statistics and Reproducibility

**Quantitation of protein levels from confocal images.** GLD-1 and GFP protein levels were quantitated in summed projections of confocal stacks using ImageJ as described[31]. Pixel intensity was measured along the distal-proximal axis of the germline using the plot profile function and a line width of 30 or 50 pixels. Intensities were copied into Excel and then transferred to Graphpad Prism (versions 9.4.1-10.2.2) and the Row Statistics function was used to calculate mean values along the distal-proximal axis. The means of both wild-type and mutants were normalized to the maximum level of protein found in the distal 100 µm of wild-type germlines and then normalized means were graphed along with 95% confidence intervals. Replicates and n for each experiment are given in figures and legends. P values were calculated in Graphpad Prism (versions 9.4.1 - 10.2.2) using a two-sided unpaired t-test assuming equal variance. P values are given for pooled data from regions of the germline indicated in each graph. Significance: *** $p < 0.001$, ** $p < 0.01$, * $p < 0.05$, ns (not significant) $p > 0.05$. Exact p values are reported in Supplementary Table 2.

**Quantitation and statistical analysis of EMSAs.** EMSA band intensities were quantitated with ImageQuant 5.2. The intensities were used to calculate the fraction of bound RNA using the formula: (intensity of bound RNA)/(intensity of bound RNA + intensity of unbound RNA). The fractions of bound RNA and FBF-2 protein concentrations were entered into GraphPad Prism 10, and data points for each replicate were fit using nonlinear regression with a model for specific binding with a Hill slope. Mean $K_d$'s and SEM were calculated in Excel. The distinct technical replicates for each experiment are reported in Fig. 2f and Table 2.

**Statistical analysis of eCLIP adjacent sites.** Peak heights were previously calculated, and significance of peak height differences was calculated in GraphPad Prism 10.0.3 using a two-sided unpaired t-test assuming equal variance. *P* values are given in figure legends.

## Reporting summary

Further information on research design is available in the Nature Portfolio Reporting Summary linked to this article.

## Data availability

Further information and requests for resources and reagents should be directed to and will be fulfilled by Traci Hall (hall4@niehs.nih.gov) and Judith Kimble (kimble@wisc.edu). Plasmids generated in this study will be deposited to Addgene. Worm strains are available from the *Caenorhabditis* stock center. Atomic coordinates and structure factors for the crystal structure generated in this study have been deposited in the Protein Data Bank under accession number 8VIV. Cryo-EM maps for the reconstructions generated in this study have been deposited in the Electron Microscopy Data Bank under the accession numbers EMD-45096 and EMD-45097. This paper analyzes existing, publicly available eCLIP data deposited at GEO under accession number GEO: GSE233561. Any additional information required to reanalyze the data reported in this paper is available from the corresponding authors upon request. Source data are provided with this paper.

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

## Acknowledgements

We thank Lars Pedersen for crystallographic and data collection support at NIEHS. Data were collected at Southeast Regional Collaborative Access Team (SER-CAT) 22-ID beamline at the Advanced Photon Source, Argonne National Laboratory. SER-CAT is supported by its member institutions, and equipment grants (S10_RR25528, S10_RR028976 and S10_OD027000) from the National Institutes of Health. Use of the Advanced Photon Source was supported by the U.S. Department of Energy, Office of Science, Office of Basic Energy Sciences, under Contract No. W-31-109-Eng-38. The authors acknowledge use of the Structural Biology Core at the National Institute of Environmental Health Sciences (ZIC ES102645). We also thank Jane Selegue, Deep Kapadia and Hezouwe Walada for technical assistance and Laura Vanderploeg for help with figures. Some nematode strains were provided by the CGC, which is funded by NIH Office of Research Infrastructure Programs (P40 OD010440). We thank our NIEHS colleagues Guang Hu and Huanchen Wang for critical reading of the manuscript. This work was supported in part by the Intramural Research Program of the National Institutes of Health, National Institute of Environmental Health Sciences [1ZIA-ES050165 to T.M.T.H., 1ZIC-ES103326 to M.J.B.] and National Institutes of Health grants [R01NS114018 to Z.T.C., R01GM50942 to M.W.; and R01GM134119 to J.K.]. This work was also supported by the National Science Foundation Graduate Research Fellowship Program to B.H.C. via DGE-1256259 and DGE-1747503; Any opinions, findings, and conclusions or recommendations expressed in this material and those of the authors and do not necessarily reflect the views of the National Science Foundation. Funding for open access charge: National Institute of Environmental Health Sciences. Some strains were provided by the CGC, which is funded by NIH Office of Research Infrastructure Programs (P40 OD010440).

## Author contributions

Protein expression and purification: C.Q., R.N.W.; X-ray crystallography: C.Q.; Protein interaction and RNA-binding analyses: C.Q.; *C. elegans* in vivo analyses: S.L.C., S.J.C.D.S., J.W.; SEC-MALS: R.C.D.; Cryo-EM: C.Q., L.B.D., V.P.D., E.G.V., M.J.B.; Bioinformatic analyses: B.H.C., T.M.T.H.; Writing and figure preparation: C.Q., S.L.C., R.N.W., Z.T.C., B.H.C., J.K., T.M.T.H.; Project management: C.Q., Z.T.C., M.J.B., S.L.C., M.W., J.K., T.M.T.H.

## Funding

## Competing interests

The authors declare no competing interests.
