## [Peer Review File · Nature Communications]

REVIEWER COMMENTS

Reviewer #1 (Remarks to the Author):

This manuscript by Qiu and colleagues investigates the molecular mechanism by which PUF RNA-binding proteins regulate germline stem cells in the nematode *Caenorhabditis elegans*. This collaborative work by the Hall and Kimble laboratories highlights the insights that can be gained by combining in vitro biophysical and structural studies with the in vivo functional analysis of stem cell behavior. In this work, they provide compelling evidence that a single LST-1 molecule can bridge two PUF-2 molecules bound to RNA to regulate the key germline *gld-1* mRNA target. This study provides a mechanistic advance for our understanding germline stem cell behavior in an important model system and will engender interest from both specialists and general readers. I found the work to be well done and clearly presented. The authors may wish to consider the following points.

1. I'm not sure I like the use of the term "higher order" to describe the PUF complex (especially in the title of the manuscript). The concern is two fold: prospective readers might think the manuscript has to do with biomolecular condensates; or readers might think the authors are talking about how RNA-binding proteins might mediate multivalent interactions with several RNA molecules.
2. I am not convinced about the authors' conclusion that translational activation of *gld-1* mRNA is affected in the FBEamFBEa*m or FBEa*mFBEbm situations (Figure 3, E-I). The concern is that translational derepression in the distal region of the gonad might increase mRNA turnover. For at least one of these cases, can the authors show that the spatial pattern of *gld-1* mRNA levels are unaffected by the 3'UTR mutations (for example using single-molecule fluorescence in situ hybridization)?
3. The binding constants measured for the formation of the higher order complexes seem rather low for mediating the potent biological effects of these molecules. Can the authors comment on this?
4. The authors discuss their in vivo results in the context of germline behavior and GLD-1 levels; however, the phenotypic endpoint seems to involve germline sex determination. This may confuse readers without additional clarification.

Reviewer #2 (Remarks to the Author):

This manuscript by Kimble, Hall, and colleagues expands upon previous work that demonstrated that PUF proteins work in concert with LST-1 to coordinate stem cell self-renewal in *C. elegans*. The current work dives into complex mechanistic questions, combining biochemical, structural, and phenotypic assays to evaluate how RNA-recognition properties and higher order complex formation can influence a critical and conserved biological pathway. The manuscript includes multiple lines

of evidence to provide a complete picture as to how RNA-binding proteins can achieve specific regulation of target mRNAs. The main advances are as follows: 1. The authors identify a cryptic FBF binding element (FBE) in the 3'UTR of *gld-1* mRNA that is required for full regulation in animals. The authors show how this new element works in concert with previously identified elements to form a higher order complex that requires two copies of FBF and one copy of LST-1 in order to achieve full activity. They also show that the c-terminus of FBF contains an auto inhibitory domain that is disrupted through competitive binding with peptide motifs from LST-1. The authors also reveal the shape of the complex using both crystallography and cryoelectron microscopy. The work is well performed and well described. The work will be of broad interest to researchers who study protein-RNA interactions, germ cell biology, and mRNA regulatory mechanisms. As such, I support publication. I do have some suggestions concerning presentation and interpretation for the authors to consider.

1. Figure 2 panel D has no shading to describe confidence intervals. Was that an oversight?
2. Figure 2 panel E through H. It would be useful to bring germline images from the supplement into the main body of the manuscript, especially for non-specialists. Is it possible to include the FBEa and FBEb data graphs in this figure? I believe they were part of a previous manuscript, but it would be nice to compare the *a/a** mutant to the *a/b* mutant side by side.
3. Could Figure 2 panel C and I be combined? I found myself trying to compare the two tables to see if specific pattern changes correlated to the sterility phenotype (or not).
4. Table1: A discussion as to why PIM B stimulates binding much more than PIM A is warranted. Also, for the ITC data, the error of the fitted parameters should be given.
5. Figure 4 and related supplemental figures. A slower mobility band is attributed to non-specific interactions between the PIM domains and the RNA. Do we know that the PIM domains bind to RNA without sequence specificity? What is the affinity? Could the PIM-RNA interaction relevant to higher order complex formation?
6. Some of the EMSA data is fit to a one-site model when the cartoon graphics of the complex would suggest a higher order model is warranted. A rationale for the use of a single site model should be presented. In cases where a single site model is used to describe a larger complex, the K_d should be reported as apparent.

Reviewer #3 (Remarks to the Author):

In this manuscript titled "A higher order PUF complex is central to regulation of *C. elegans* germline stem cells" Qui C. et al uncover two mechanisms of RNA-dependent regulation of germline stem cell differentiation in nematodes by the FBF-2 and LST-1 proteins. First, they combine isothermal calorimetry and SEC-MALLS experiments to demonstrate that LST-1 binding to FBF-2 triggers the

formation of a 1:2 (LST-1:FBF-2) complex to enhance RNA recognition by FBF-2. They strategically used mutations in LST-1 to further show that its PIM-B domain plays a stronger contribution than PIM-A but both are required for higher oligomeric assembly. Second, they demonstrate the presence of a new RNA binding motif for FBF-2 in the RNA Gld-1, termed FBEa*. Using non-denaturing PAGE, the authors quantify the binding of FBF-2 to FBEa*. To further validate their proposal, they performed extensive mutational analysis in *C. elegans* revealing that FBEa* is important for fertility and the control of Gld-1 levels. They even solve the structure of FBF-2 bound to this new 'non-canonical' FBEa*. Overall, the work is of high quality with proper controls and the authors convincingly support their statements. They elegantly combined a diverse set of methods in an interdisciplinary approach to reveal a previously unknown mode of regulation of the Gld-1 transcript by FBF-2. Their work is significant as they propose a new mechanism for FBF-2 function in RNA post-transcriptional regulation that involves higher oligomeric protein assembly and the interplay of additional RNA binding sites. This mechanism has implications for the large family of PUF proteins (to which FBF-2 belongs) and will likely motivate others to test the prevalence or not of this mechanism. Together, their work will be of high interest to the broad RNA, structural and stem cell communities and is a strong fit for Nature Communications.

I am not equipped to fully assess the quality of the work done in *C. elegans* but I have a major concern about the interpretation of their cryo-EM data that should be addressed by the authors before approval for publication.

- The biggest concern I have is the quality of the cryo-EM reconstruction. The authors used cryo-EM to visualize the novel 2:1:1 FBF-2:LST-1:RNA complex they report. The quality of their map is limited (6.4 Angs global resolution) and I am not convinced they can see the RNA feature they state they are seeing. Moreover, the authors identify a subset of particles that show two FBF-2 molecules and speculate that they are the 2:1:1 complex they've assembled and purified by size exclusion chromatography. While it is possible that the two FBF-2 molecules seen in proximity are from the 2:1:1 complex, it is equally possible that these are particles closely packed due to crowding effect during grid preparation. In other words, it is hard to say if the 2:1:1 complex remained stable during grid preparation. This issue might be alleviated if a dataset is collected on a more modern/powerful microscope since the dataset comes from a Talos Arctica 200 kEV with a K2 summit direct detection camera – challenging to obtain good quality data for such a small complex on such a microscope. Alternatively, the authors can compare cryo-EM datasets of FBF-2:LST-1 in the absence of RNA and free FBF-2 to validate the presence or not of these “clustered FBF-2” particles. As it stands, the model for the 2:1:1 complex based on cryo-EM could be over-interpreted.
- The authors solved the crystal structure of FBF-2 RNA binding domain bound to FBEa*. This is a challenging feat, yet the authors did not sufficiently discuss insights gained from this new structure. For instance, does this structure rationalize the 6-fold weaker binding of FBF-2 to FBEa* compared to FBEa? Are there differences on the protein when binding FBEa vs FBEa*? How is U9 in FBEa* recognized compared to A9 in FBEa? From Fig 2 it seems that FBEa covers a larger surface of FBF-2 and extends towards more repeats compared to FBEa*. More analysis and discussion of their newly solved crystal structure would be greatly beneficial.

Additional minor comments:

- Fig S1C shows strong heat during the ITC titration (~ -1 uW) yet the authors interpret this as non-binding. This contrasts with the result in Fig S1F where there is clearly no binding with minimal heat detected. The authors should address the source of this strong background heat in Fig S1C when both PIM-A and PIM-B are mutated and titrated with RBD+CT. While I do not disagree with their overall statement that mutating both PIM motifs in LST-1 blocks the interaction with FBF-2, I worry that the strong residual heat during the titration might be caused by buffer mismatching or potential interactions with CT since such heat is not seen in its absence.
- It seems their X-ray crystal structure is of higher quality than 2.29 Angs given a mean $I/\sigma(I)$ of 2 in the highest resolution shell. Why did the author not process their dataset to its highest possible resolution? The authors should add the $cc1/2$ value in Supplementary Table 1.
- Line 23 replace 'and' with 'in'

RESPONSE TO REVIEWER COMMENTS

Reviewer #1 (Remarks to the Author):

This manuscript by Qiu and colleagues investigates the molecular mechanism by which PUF RNA-binding proteins regulate germline stem cells in the nematode *Caenorhabditis elegans*. This collaborative work by the Hall and Kimble laboratories highlights the insights that can be gained by combining in vitro biophysical and structural studies with the in vivo functional analysis of stem cell behavior. In this work, they provide compelling evidence that a single LST-1 molecule can bridge two PUF-2 molecules bound to RNA to regulate the key germline *gld-1* mRNA target. This study provides a mechanistic advance for our understanding germline stem cell behavior in an important model system and will engender interest from both specialists and general readers. I found the work to be well done and clearly presented. The authors may wish to consider the following points.

1. I'm not sure I like the use of the term "higher order" to describe the PUF complex (especially in the title of the manuscript). The concern is two fold: prospective readers might think the manuscript has to do with biomolecular condensates; or readers might think the authors are talking about how RNA-binding proteins might mediate multivalent interactions with several RNA molecules.

Author response: We appreciate Reviewer 1's concern, however, the other reviewers did not voice a similar concern. The term "higher order" is succinct, and we hope by clearly defining it, we will avoid confusion with other possible uses.

The original manuscript explained our use of "a higher order complex" at numerous critical junctures. However, the term was missing from the Abstract, which was an oversight. The revised manuscript now adds the term to the Abstract to make sure the term is defined uniformly throughout.

Abstract: Lines 9-12, with change on line 11 bolded and underlined: *"Discoveries of the LST-1-FBF-2 ternary complex, the *gld-1* adjacent FBEs, and their in vivo significance predict an expanded regulatory repertoire of different assemblies of PUF-partner-**RNA higher order complexes** in nematode GSCs."*

We also more directly define "higher order PUF complexes" on pages 4-5 (lines 84-85) as "a ternary complex composed of two PUF proteins linked by a multivalent partner protein that binds to two RNA regulatory elements."

2. I am not convinced about the authors' conclusion that translational activation of *gld-1* mRNA is affected in the FBE^{am}FBE^{a*m} or FBE^{a*m}FBE^{bm} situations (Figure 3, E-I). The concern is that translational derepression in the distal region of the gonad might increase mRNA turnover. For at least one of these cases, can the authors show that the spatial pattern of *gld-1* mRNA levels are unaffected by the 3'UTR mutations (for example using single-molecule fluorescence in situ hybridization)?

Author response: Data presented in this paper show unequivocally that distal and proximal effects on GLD-1 levels are not coupled (Figs. 3e, 3f, and 3j), which makes the concern raised by Reviewer 1 highly unlikely. Graphs in Figs. 3e, 3f, and 3j demonstrate significant distal derepression but no accompanying decrease to proximal activation. Therefore, the

idea that the proximal lowering of GLD-1 might be a downstream consequence of distal depression does not hold water.

To avoid misunderstanding, the revised manuscript now explicitly points out how graphs of Figs. 3e, f and j argue against this idea. That addition is:

Page 9, lines 223-226. *“One concern might have been that lowered proximal GLD-1 represents a downstream consequence of distal derepression rather than a defect in proximal activation. However, distal derepression is not coupled to proximal deactivation (Fig. 3e, f, and j).”*

3. The binding constants measured for the formation of the higher order complexes seem rather low for mediating the potent biological effects of these molecules. Can the authors comment on this?

Author response: *In vitro* binding constants are dependent on experimental conditions and the technique used, and therefore relative values are most instructive. For example, we used tRNA to limit non-specific binding in our reactions for the EMSAs, but without tRNA, the affinity is 10-fold tighter. We observed the same 10-fold higher affinity binding for previous experiments using surface plasmon resonance, which use relatively simple buffers lacking tRNA, BSA, or Tween.

With this reasoning in mind, the 109 nM *in vitro* binding constant for the FBF-2 RBD+CT/LST-1¹⁹⁻⁹⁸ interaction with 29-nt *gld-1* FBEa-FBEa* RNA is not especially weak. This value is in the same ballpark as other *in vitro* measurements for FBF-2: 68 nM for FBF-2 RBD+CT/LST-1¹⁹⁻⁵⁰ binding to a 14-nt FBEa RNA by EMSA (Qiu 2023 Nature Comm), 70 nM for FBF-2 RBD to the 14-nt FBEa RNA by EMSA (Qiu 2023 Nature Comm), and 110 nM for FBF-2 RBD to a Cy5-labeled 14-nt FBEa RNA by MicroScale Thermophoresis (Qiu 2022 NAR).

We thank Reviewer 1 for raising this issue since others will likely have the same question. The revised manuscript includes a new sentence to address this concern explicitly:

Page 10, lines 259-260: *“These overall binding affinities are similar to those for FBF-2 RBD+CT (334 nM), RBD (70 nM), and L610A (82 nM) binding to a 14-nt FBEa RNA⁹.”*

4. The authors discuss their *in vivo* results in the context of germline behavior and GLD-1 levels; however, the phenotypic endpoint seems to involve germline sex determination. This may confuse readers without additional clarification.

Author response: We thank Reviewer 1 for pointing this out. The revised manuscript adds a sentence to address this issue.

Page 8, lines 193-195. *“Failure of the sperm/oocyte switch was also seen when both canonical elements, FBEa and FBEb, were mutated (Fig. 3c, line 7)³¹.”* (Carrick et al., 2024)

David Greenstein

Reviewer #2 (Remarks to the Author):

This manuscript by Kimble, Hall, and colleagues expands upon previous work that demonstrated that PUF proteins work in concert with LST-1 to coordinate stem cell self-renewal in *C. elegans*. The current work dives into complex mechanistic questions, combining biochemical, structural, and phenotypic assays to evaluate how RNA-recognition properties and higher order complex formation can influence a critical and conserved biological pathway. The manuscript includes multiple lines of evidence to provide a complete picture as to how RNA-binding proteins can achieve specific regulation of target mRNAs. The main advances are as follows: 1. The authors identify a cryptic FBF binding element (FBE) in the 3'UTR of *gld-1* mRNA that is required for full regulation in animals. The authors show how this new element works in concert with previously identified elements to form a higher order complex that requires two copies of FBF and one copy of LST-1 in order to achieve full activity. They also show that the c-terminus of FBF contains an auto inhibitory domain that is disrupted through competitive binding with peptide motifs from LST-1. The authors also reveal the shape of the complex using both crystallography and cryoelectron microscopy. The work is well performed and well described. The work will be of broad interest to researchers who study protein-RNA interactions, germ cell biology, and mRNA regulatory mechanisms. As such, I support publication. I do have some suggestions concerning presentation and interpretation for the authors to consider.

1. Figure 2 panel D has no shading to describe confidence intervals. Was that an oversight?

Author response: We first note that we believe the reviewer is referring to Fig. 3 in points 1-3. We are confused that Reviewer 2 could not view the gray confidence interval shading for the wt control in Fig. 3d.

2. Figure 2 panel E through H. It would be useful to bring germline images from the supplement into the main body of the manuscript, especially for non-specialists. Is it possible to include the FBEa and FBEb data graphs in this figure? I believe they were part of a previous manuscript, but it would be nice to compare the *a/a** mutant to the *a/b* mutant side by side.

Author response: As requested, we generated a revised Fig. 3 that adds data graphs for mutations in FBEa and FBEb (new Fig. 3f and 3g). Also as requested, the revised Fig. 3 adds a representative image of GLD-1 staining (new Fig. 3d). However, with respect, we suggest one image is sufficient to get the idea across.

3. Could Figure 2 panel C and I be combined? I found myself trying to compare the two tables to see if specific pattern changes correlated to the sterility phenotype (or not).

Author response: As requested, the revised Fig. 3 creates a revised panel, Fig. 3c, that combines previous panels c and i.

4. Table1: A discussion as to why PIM B stimulates binding much more than PIM A is warranted. Also, for the ITC data, the error of the fitted parameters should be given.

Author response: We added an explanation about why PIM-B binds more tightly than PIM-A, referring to our previous work. We had demonstrated that residues flanking the KxxL motifs in the PIMs produce the differences in binding affinity (Qiu 2022 NAR).

Page 6, lines 129-131. *"We previously demonstrated with shorter peptides containing a single LST-1 PIM that PIM-B binds with higher affinity to FBF-2 than PIM-A, and this tighter binding of PIM-B is due to amino acid residues flanking the KxxL motifs³⁰."*

We also added the errors for the ITC values in Table 1.

5. Figure 4 and related supplemental figures. A slower mobility band is attributed to non-specific interactions between the PIM domains and the RNA. Do we know that the PIM domains bind to RNA without sequence specificity? What is the affinity? Could the PIM-RNA interaction relevant to higher order complex formation?

Author response: We consistently observe intermediate bands just above unbound RNA in assays that include LST-1, particularly PIM-B. A control experiment with LST-1¹⁹⁻⁹⁸ and FBEa-FBEa* RNA, omitting FBF-2, revealed a band just above unbound RNA at the concentration of LST-1¹⁹⁻⁹⁸ used in our EMSAs (10 μ M), allowing us to attribute the band to LST-1¹⁹⁻⁹⁸ and RNA.

In response to Reviewer 2's questions, we performed two additional control experiments to address the possibility of sequence specificity. (1) A control experiment with serial dilutions of LST-1¹⁹⁻⁹⁸ added to FBEa^m-FBEa^{*m} RNA, where both FBEs are mutated, also showed a band just above unbound RNA that became prominent at \sim 3 μ M LST-1¹⁹⁻⁹⁸. Aggregates that are trapped in the wells occur at the highest concentrations. (2) We performed an EMSA with serial dilutions of FBF-2 RBD+CT and a constant 10 μ M LST-1¹⁹⁻⁹⁸ added to FBEa^m-FBEa^{*m} RNA. This is equivalent to Fig. 4d, but with the mutated RNA. We observed the intermediate band in this experiment with the mutated RNA. Based on the association with mutated RNA, we conclude that LST-1¹⁹⁻⁹⁸ interaction with RNA is not sequence specific. We could not determine an affinity of the interaction due to the aggregation at the highest concentrations, but it would be in the μ M range. We include these gels in the source data file.

Since the interaction appears to be non-specific and we have no evidence that LST-1 interacts with the RNA in the complex, we find it unlikely that it is relevant to higher order complex formation.

6. Some of the EMSA data is fit to a one-site model when the cartoon graphics of the complex would suggest a higher order model is warranted. A rationale for the use of a single site model should be presented. In cases where a single site model is used to describe a larger complex, the K_d should be reported as apparent.

Author response: This is a good point. All EMSA data were fit using nonlinear regression with a model for specific binding with a Hill slope. The Hill slope values were greater than 1

for both the FBEa* RNA or FBEa-FBEa* RNA. For experiments with FBEa-FBEa* RNAs, fitting with a two-site model failed, so we used the same model for specific binding with a Hill slope for all experiments. We added these details to the Methods (page 23, lines 669-673) and report $K_{D,app}$ values where appropriate.

Reviewer #3 (Remarks to the Author):

In this manuscript titled “A higher order PUF complex is central to regulation of *C. elegans* germline stem cells” Qui C. et al uncover two mechanisms of RNA-dependent regulation of germline stem cell differentiation in nematodes by the FBF-2 and LST-1 proteins. First, they combine isothermal calorimetry and SEC-MALLS experiments to demonstrate that LST-1 binding to FBF-2 triggers the formation of a 1:2 (LST-1:FBF-2) complex to enhance RNA recognition by FBF-2. They strategically used mutations in LST-1 to further show that its PIM-B domain plays a stronger contribution than PIM-A but both are required for higher oligomeric assembly. Second, they demonstrate the presence of a new RNA binding motif for FBF-2 in the RNA *Gld-1*, termed FBEa*. Using non-denaturing PAGE, the authors quantify the binding of FBF-2 to FBEa*. To further validate their proposal, they performed extensive mutational analysis in *C. elegans* revealing that FBEa* is important for fertility and the control of *Gld-1* levels. They even solve the structure of FBF-2 bound to this new ‘non-canonical’ FBEa*. Overall, the work is of high quality with proper controls and the authors convincingly support their statements. They elegantly combined a diverse set of methods in an interdisciplinary approach to reveal a previously unknown mode of regulation of the *Gld-1* transcript by FBF-2. Their work is significant as they propose a new mechanism for FBF-2 function in RNA post-transcriptional regulation that involves higher oligomeric protein assembly and the interplay of additional RNA binding sites. This mechanism has implications for the large family of PUF proteins (to which FBF-2 belongs) and will likely motivate others to test the prevalence or not of this mechanism. Together, their work will be of high interest to the broad RNA, structural and stem cell communities and is a strong fit for Nature Communications.

I am not equipped to fully assess the quality of the work done in *C. elegans* but I have a major concern about the interpretation of their cryo-EM data that should be addressed by the authors before approval for publication.

- The biggest concern I have is the quality of the cryo-EM reconstruction. The authors used cryo-EM to visualize the novel 2:1:1 FBF-2:LST-1:RNA complex they report. The quality of their map is limited (6.4 Angs global resolution) and I am not convinced they can see the RNA feature they state they are seeing. Moreover, the authors identify a subset of particles that show two FBF-2 molecules and speculate that they are the 2:1:1 complex they’ve assembled and purified by size exclusion chromatography. While it is possible that the two FBF-2 molecules seen in proximity are from the 2:1:1 complex, it is equally possible that these are particles closely packed due to crowding effect during grid preparation. In other words, it is hard to say if the 2:1:1 complex remained stable during grid preparation. This issue might be alleviated if a dataset is collected on a more modern/powerful microscope since the dataset comes from a Talos Arctica 200 kEV with a K2 summit direct detection camera – challenging to obtain good quality data for such a small complex on such a microscope. Alternatively, the authors can compare cryo-EM datasets of FBF-2:LST-1 in the absence of RNA and free FBF-2 to validate the presence or not of these “clustered FBF-2” particles. As it stands, the model for the 2:1:1 complex based on cryo-EM could be over-interpreted.

Author response: We agree that it is challenging to obtain good quality data for such a small complex using any microscope. Using 200 keV and K2 DED cameras can provide sub-3 Å resolution structures for less than 200 kDa molecular weight-sized complexes (Herzik, M., Wu, M. & Lander, G. Achieving better-than-3-Å resolution by single-particle cryo-EM at 200 keV. *Nat Methods* 14, 1075–1078 (2017). <https://doi.org/10.1038/nmeth.4461>). For our sample we believe that the microscope and camera are not limiting, as we are at 6.4 Å resolution. The sample and dataset reported here yielded the best reconstruction from our many attempts. We recently attempted to improve the reconstruction with the Blush denoise tool in Relion (Kirmanius, et al. 2024, *Nature Methods* 21:1216-1121), but the maps did not improve.

Although our overall conclusion about the function of the FBF-2/LST-1/RNA higher order complex does not depend upon a structure, our goal in presenting the cryo-EM model is to visualize the complex, recognizing the limitations in resolution. We reworded the subtitle (page 11, line 288, “Cryo-EM **visualizes** a higher-order complex of FBF-2/LST-1/RNA”) and the description of the structures on pages 11-12, lines 306-317 to better communicate this (changes indicated in bold below).

Pages 11-12, lines 306-317. “After *ab initio* reconstruction and refinement in CryoSparc⁴⁰, we obtained **two maps: a map of a single FBF-2 molecule displaying rod-like densities for individual helices and densities for RNA bases at 4.4 Å resolution (Fig. 5b, Supplementary Fig. 6, v-1) and a map with two well-resolved FBF-2 molecules with apparent densities for RNA bases at 6.4 Å resolution (Fig. 5c, Supplementary Fig. 6, v-2)**. We docked models of FBF-2/LST-1/RNA complexes in the final 3D reconstructions (Fig. 5b, c; right). However, the resolution was limited **and we did not fit the models further**. The 35-residue linker between the LST-1 PIMs as well as the three nucleotides between FBEa and FBEa* are expected to introduce structural flexibility in the relative arrangements of the two FBF-2 molecules. Nonetheless, the cryo-EM reconstruction **provides a visual picture of how FBF-2, LST-1¹⁹⁻⁹⁸, and FBEa-FBEa* RNA can form a quaternary complex with two FBF-2 molecules bound to LST-1 and a single RNA.**”

We also adjusted the orientations of the model in Fig. 5c to better show that there is density in the map for the RNA.

The reviewer’s idea that the 2:1:1 complex cryo-EM reconstruction could arise from adjacent FBF-2 molecules, not necessarily bound to RNA or LST-1, is one that we also considered. We concluded it was unlikely for the following reasons:

1-In order to obtain a cryo-EM reconstruction with two well resolved FBF-2 molecules, the two FBF-2 molecules must be arranged in the same orientation relative to each other. If not, one molecule would appear with clear density but the other would be blurred because it is in different relative orientations to the first FBF-2 and the 2D classes would not converge. We built our 2:1:1 complex reconstruction from ~9% of the total particles. It is difficult to imagine that random crowding or clustering of single FBF-2 molecules would produce 9% of FBF-2 pairs in one relative conformation.

2-We mapped the particles used in the reconstruction back to the micrographs and assessed the density of particles. In these representative examples of denoised images, the relatively low particle density can be observed. We include the left panel in Suppl. Fig. 5c.

- The authors solved the crystal structure of FBF-2 RNA binding domain bound to FBEa*. This is a challenging feat, yet the authors did not sufficiently discuss insights gained from this new structure. For instance, does this structure rationalize the 6-fold weaker binding of FBF-2 to FBEa* compared to FBEa? Are there differences on the protein when binding FBEa vs FBEa*? How is U9 in FBEa* recognized compared to A9 in FBEa? From Fig 2 it seems that FBEa covers a larger surface of FBF-2 and extends towards more repeats compared to FBEa*. More analysis and discussion of their newly solved crystal structure would be greatly beneficial.

Author response: Thank you to the reviewer for recognizing the challenge of determining a new crystal structure. We were concise in our description of the structure as the length of the manuscript is quite long, but it appears that we were too concise based on the questions. We revised this section of the manuscript on page 7 (lines 157-160) and added panels to Suppl. Fig. 2.

Page 7, lines 157-160. *“The FBF-2 RBD structure changes little (**Supplementary Fig. 2a**, RMSD = 0.37 Å over 2,744 atoms). The RNA sequences align in the 5′ and 3′ regions, and the side chains that contact the RNA bases in each structure are equivalent and positioned similarly (**Supplementary Fig. 2a-c**).”*

1-The new Suppl. Fig. 2a with a superposition of the structures of FBF-2 bound to FBEa and FBEa* RNA should illustrate that there is very little difference in the FBF-2 RBD structures when bound to the two RNAs (RMSD for FBF-2 = 0.37 Å over 2,744 atoms).

2-The new Suppl. Figs. 2b and 2c should illustrate how the FBF-2 repeats align relative to the two RNA sequences. We hope this makes it obvious that the sequences bound by repeats R2-R8′ are the same except U4 in FBEa*. This leads to the same contacts being made by FBF-2 to the two different RNAs.

3-The new Suppl. Figs. 2b and 2c should also more clearly illustrate that the FBEa* RNA in the crystal structure lacks an equivalent nucleotide to A9 in the FBEa RNA (the RNAs have the same number of flanking nucleotides for binding experiments). FBF-2 recognition at the

position occupied by FBEa A9 is not sequence specific (Opperman 2005 NSMB, Wang 2009 PNAS, Prasad 2016 RNA, Carrick 2024 Dev Cell). Therefore, the reviewer is correct that the RNA-binding surface for FBEa on FBF-2 appears larger than for FBEa*, but it is due to the additional 3' nucleotide rather than a change in FBF-2.

4-Finally, the structure, though beautiful, does not explain why FBF-2 binds to FBEa* with lower affinity. This is not uncommon that crystal structures of PUF proteins reveal apparently similar contacts but different binding affinities. The structure is a static picture and does not account for factors like the energy required to flip the U4 base of FBEa* or the destabilization that the flipped base may cause. We added a sentence to page 7 to address this point.

Page 7, lines 168-170. *“The difference in binding affinity may therefore be due to other factors such as additional energy needed to flip the U4 base of FBEa* or destabilization of binding caused by the flipped base.”*

Additional minor comments:

- Fig S1C shows strong heat during the ITC titration (~-1 uW) yet the authors interpret this as non-binding. This contrasts with the result in Fig S1F where there is clearly no binding with minimal heat detected. The authors should address the source of this strong background heat in Fig S1C when both PIM-A and PIM-B are mutated and titrated with RBD+CT. While I do not disagree with their overall statement that mutating both PIM motifs in LST-1 blocks the interaction with FBF-2, I worry that the strong residual heat during the titration might be caused by buffer mismatching or potential interactions with CT since such heat is not seen in its absence.

Author response: This is a helpful observation. We did carefully match buffers in our ITC assays, so that is an unlikely cause of the residual heat. We agree that the high residual heat is observed when the experiment includes FBF-2 RBD+CT but not FBF-2 RBD, and therefore potential interactions with the FBF-2 CT are the likely cause. We added an explanatory note to the legend for Suppl. Fig. 1c.

Supplementary Information, page 8. *“Experiments that include FBF-2 RBD+CT (a-c) show residual heat that was not observed for those including FBF-2 RBD (d-f), suggesting this heat may be attributed to potential interactions with the FBF-2 CT.”*

- It seems their X-ray crystal structure is of higher quality than 2.29 Å given a mean $I/\sigma(I)$ of 2 in the highest resolution shell. Why did the author not process their dataset to its highest possible resolution? The authors should add the $cc1/2$ value in Supplementary Table 1.

Author response: This is a good point. We determined the crystal structure a few years ago using the criteria of $I/\sigma(I) > 2$ for resolution cutoff. We have reprocessed the data to current standards. The resolution improved from 2.29 Å to 2.20 Å, with $cc1/2 \sim 0.50$ in the last shell. The statistics are updated in Supplementary Table 1.

- Line 23 replace 'and' with 'in'

Author response: Done

REVIEWERS' COMMENTS

Reviewer #1 (Remarks to the Author):

The authors have done a nice job addressing all the critiques previously provided. I have no additional comments.

Reviewer #2 (Remarks to the Author):

I am satisfied by the reviewers responses to my questions / suggestions and support publication without further delay.

Reviewer #3 (Remarks to the Author):

The authors did a good job addressing all comments raised by the three reviewers. I have no more concerns and the structural work is solid and of high interest to the field.